# MAPO: Momentum-Aware Policy Optimization

## Abstract

Reinforcement Learning with Verifiable Rewards (RLVR) has emerged as a promising framework to enhance the reasoning capabilities of Large Language Models (LLMs), yet the samples from the policy model are not fully exploited during training. We propose **M**omentum-**A**ware **P**olicy **O**ptimization (**MAPO**), a critic-free, drop-in framework that preserves the simplicity of GRPO while improving exploration and stability. MAPO introduces (i) a Momentum Group Baseline that yields non-vanishing learning signals under group-standardized rewards; (ii) confidence-based prioritized replay that reuses verified successes to increase sample efficiency; and (iii) entropy-weighted token updates that concentrate gradient mass on uncertain decision points. Evaluated on math reasoning benchmarks, MAPO outperforms strong baselines—including GRPO and DAPO—in best-of-$N$ accuracy (pass@$N$), demonstrating superior exploration and discovery of correct reasoning trajectories. Ablation studies attribute the primary gains to the momentum advantage, which reduces the steps required to reach the target, alleviates stalls on homogeneous reward groups, and reduces across-seed variance. The replay and entropy components provide complementary improvements in sample utilization and gradient allocation. Overall, MAPO achieves target performance in fewer optimization steps while maintaining training stability, offering a practical enhancement to group-based RLVR methods.

## 1 Introduction

Large language models have demonstrated remarkable capabilities in complex reasoning tasks, yet achieving reliable performance in domains requiring precise logical steps remains challenging(Jaech et al., 2024; Guo et al., 2025; Bai et al., 2023). The key insight driving recent progress is that these domains offer access to automated verification: unlike subjective human preferences, mathematical proofs and code execution provide objective, binary feedback that can guide model improvement at scale(Lightman et al., 2024).

Reinforcement learning with verifiable rewards (RLVR) leverages these programmatic signals to guide policy optimization (Mroueh, 2025; Su et al., 2025). However, two fundamental obstacles persist in practice. First, low reward variance emerges when models uniformly succeed or fail on similar problems, causing normalized advantages to collapse and weakening policy gradients. Both empirical and theoretical analyses demonstrate that small reward standard deviation leads to flat objectives that stall learning progress (Razin et al., 2025). While subsequent remedies—including variance-boosting heuristics and adaptive baselines—provide partial relief, they fail to address the underlying pathology (Li et al., 2025; Wang et al., 2025a). Second, token-level credit assignment presents an inherent challenge: sequence-level rewards are distributed uniformly across all tokens despite only a subset being causally critical to the outcome, particularly in extended chains of reasoning.

We introduce **M**omentum-**A**ware **P**olicy **O**ptimization (**MAPO**), a framework that addresses these challenges through three key innovations: (1)**Momentum Group Baseline** maintains learning signals even when current samples show low reward variance by incorporating historical context, (2) **Confidence-Based Prioritized Replay** preserves valuable successful trajectories for repeated learning while preventing regression, and (3) **Entropy-Weighted Token Updates** allocates gradient updates proportionally to local uncertainty, focusing learning on critical decision points.

Extensive evaluation on mathematical reasoning benchmarks demonstrates that MAPO consistently outperforms strong baselines including GRPO and DAPO across both pass@1 and pass@N metrics. Ablation studies reveal that the momentum baseline provides the primary performance gains, while replay and entropy weighting offer complementary improvements in sample efficiency and gradient allocation. These results establish MAPO as an effective drop-in enhancement for verifiable reward settings.

## 2 RELATED WORK

**Reinforcement Learning for LLM Alignment.** The dominant paradigm for LLM alignment follows reinforcement learning from human feedback (RLHF), typically implemented via PPO with learned value functions and KL penalties to reference models (Ziegler et al., 2019; Stiennon et al., 2020; Ouyang et al., 2022). While effective, PPO's critic dependence and token-level clipping mechanisms become brittle when scaled to large models (Schulman et al., 2017). This has motivated research into reinforcement learning with verifiable rewards (RLVR), where automated checkers provide objective signals for domains like mathematics and coding (Lightman et al., 2024; Yu et al., 2023; Setlur et al., 2024).

**Group-Based Policy Optimization.** To eliminate critic dependencies, group-based methods sample multiple completions per prompt and use group statistics to construct advantages. Group Relative Policy Optimization (GRPO) (Shao et al., 2024) exemplifies this approach: for prompt $x$, the policy $\pi_{\theta_{\text{old}}}$ samples a group of $G$ responses $\{y_i\}_{i=1}^{G}$ with sequence rewards $\{r_i\}_{i=1}^{G}$. Let the group mean and standard deviation be

$$\mu = \frac{1}{G}\sum_{j=1}^{G} r_j, \qquad \sigma = \sqrt{\frac{1}{G}\sum_{j=1}^{G}(r_j - \mu)^2}.$$

GRPO assigns the same scalar advantage to every token of response $y_i$

$$\hat{A}_{i,t} = \frac{r_i - \mu}{\sigma + \varepsilon}, \qquad t = 1, \ldots, |y_i|, \tag{1}$$

where $\varepsilon > 0$ ensures numerical stability. With PPO-style clipping and KL regularization, this eliminates value function learning while proving effective for mathematical reasoning tasks.

However, GRPO's group normalization suffers when reward variance is low: if all samples succeed or fail similarly,$\sigma \approx 0$ and advantages vanish, stalling learning. Additionally, the group-normalized advantage depends on empirical statistics from finite groups of size $G$ and we quantify how $G$ affects estimator bias and reliability under Bernoulli rewards in Appendix B. This analysis explains why momentum baselines and prioritized replay mitigate gradient collapse when groups are nearly homogeneous.

**Extensions and Alternatives.** Recent scaling efforts emphasize token-level refinements: DAPO decouples clipping and employs dynamic sampling to stabilize long-CoT training (Yu et al., 2025), while Dr.,GRPO corrects bias in GRPO to improve token efficiency (Liu et al., 2025). Complementary work shows that high-entropy minority tokens disproportionately drive RLVR gains (Wang et al., 2025b). Experience replay adaptations include RLEP, which collects verified trajectories and replays high-quality successes (Zhang et al., 2025), and off-policy corrections that reduce PPO complexity (Ahmadian et al., 2024). Our confidence-based prioritized replay operates online and is conceptually aligned with prioritized experience replay from deep RL (Schaul et al., 2016) and recent LLM replay variants (Dou et al., 2025). Preference-optimization methods like DPO and KTO offer stable alternatives (Rafailov et al., 2023; Ethayarajh et al., 2024), but cannot directly exploit automated verifiers, limiting RLVR applicability.

## 3 METHOD

In this section, we introduce our method, MAPO, which augments the group-based policy optimization with momentum-based baselines, prioritized replay, and entropy-guided updates. We index outer optimization iterations by $k$, within groups by $i$, and token positions by $t$.

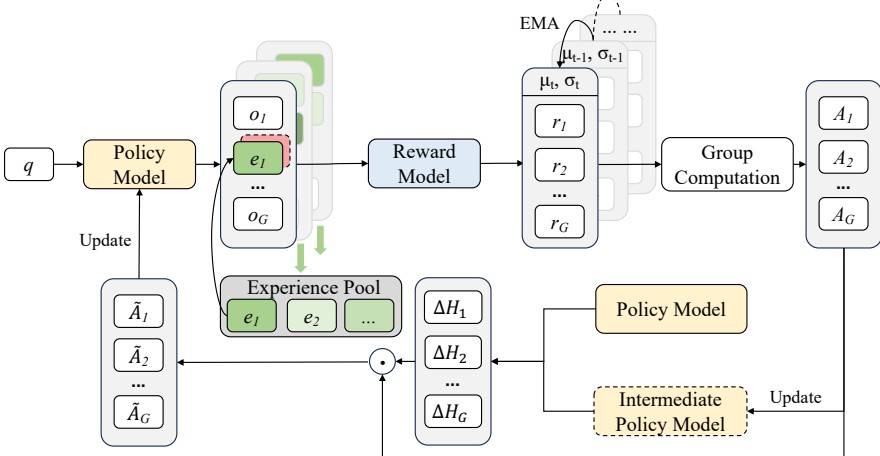

Figure 1: **MAPO architecture and workflow.** Given prompt $q$, the policy model generates candidate responses evaluated by a reward model. MAPO addresses three key challenges in group-based RLVR: (1) sample inefficiency via confidence-based experience replay of high-quality trajectories $\{e_1, e_2, \ldots\}$, (2) vanishing advantages under uniform rewards via Momentum Group Baseline $\{\mu_{t-1}, \sigma_{t-1}\}$, and (3) suboptimal credit assignment via entropyguided advantage weighting $\{\Delta H_1, \Delta H_2, \ldots, \Delta H_G\}$. The method computes group advantages $\{A_1, A_2, \ldots, A_G\}$, applies entropy-based token reweighting for stable policy updates.

### 3.1 MOMENTUM ADVANTAGE WITH MOMENTUM GROUP BASELINE

Group-based RLVR suffers from vanishing advantages when batch rewards are homogeneous, which leads to the elimination of gradient signals. This occurs frequently as models consistently solve easy prompts or fail uniformly on hard ones. We address this issue with a momentum-based baseline that carries information from past batches.

For each trajectory $i$, we maintain an *exponential moving average* (EMA) baseline

$$b_i^{(k)} = \alpha\,\mu_{-i}^{(k)} + (1-\alpha)\,b_i^{(k-1)}, \qquad b_i^{(0)} = \mu_{-i}^{(0)}. \tag{2}$$

where $\mu_{-i}^{(k)}$ denotes the group mean reward at iteration $k$ excluding trajectory $i$ and $\alpha \in (0, 1]$ controls the decay rate. The momentum advantage is

$$A_i^{(k)} = \frac{r_i^{(k)} - b_i^{(k)}}{\sigma^{(k)} + \varepsilon} + \eta\,\Delta b_i^{(k)}, \tag{3}$$

where $\Delta b_i^{(k)} := b_i^{(k)} - b_i^{(k-1)}$ is the baseline change and $\eta \geq 0$ is a small weighting hyperparameter. The first term in $A_i^{(k)}$ generalizes the advantage estimation in GRPO by using the EMA baseline $b_i^{(k)}$. The second term $\eta\Delta b_i^{(k)}$ rewards positive baseline drift (if the group's performance is improving over time, $\Delta b > 0$, it boosts the advantage) and penalizes regressions ($\Delta b < 0$ yields a negative term). Intuitively, this momentum term pushes the policy to continually outperform its historical average for that prompt, thus providing a learning signal even in batches where $r_i \approx \mu$.

**Bias–variance properties.** Within a group, let $r_j^{(k)} \stackrel{\text{i.i.d.}}{\sim} \text{Bernoulli}(p_k)$ and denote the leave-one-out mean and std by $(\mu_{-i}^{(k)}, \sigma^{(k)})$ with $\sigma_\varepsilon^{(k)} := \sigma^{(k)} + \varepsilon$. The group-normalized term $(r_i^{(k)} - \mu_{-i}^{(k)})/\sigma_\varepsilon^{(k)}$ is approximately unbiased, with residual bias of order $O(G^{-1})$ due to the random denominator; its variance is also $O(G^{-1})$ under non-degenerate rewards.

Introducing the EMA baseline $b_i^{(k)} = \alpha\mu_{-i}^{(k)} + (1-\alpha)b_i^{(k-1)}$ yields geometric bias decay in $k$ (as $b_i^{(k-1)} \to p_k$) and preserves the $O(G^{-1})$ variance scaling. For the momentum-augmented advantage

$$A_i^{(k)} = \frac{r_i^{(k)} - b_i^{(k)}}{\sigma_\varepsilon^{(k)}} + \eta\,\widetilde{\Delta b}_i^{(k)}, \qquad \widetilde{\Delta b}_i^{(k)} := \Delta b_i^{(k)} - \mathbb{E}\left[\Delta b_i^{(k)}\right], \quad \Delta b_i^{(k)} := b_i^{(k)} - b_i^{(k-1)},$$

a standard variance decomposition shows that, for the empirically typical negative covariance between the two centered terms, some $\eta \in [0, 1]$ achieves net variance reduction relative to the non-momentum case. Formal expansions for the finite-$G$ bias and the full variance bound appear in Appendix B, and Appendix C respectively.

**Non-vanishing signal under uniform rewards.** When all responses in a group share the same reward $r_j^{(k)} \equiv c \in \{0, 1\}$, GRPO collapses to a null update because $\mu_{-i}^{(k)} = c$ and $\sigma^{(k)} = 0$, hence $(c - \mu_{-i}^{(k)})/\sigma_\varepsilon^{(k)} = 0$ (with stabilization). In contrast, MAPO maintains a bounded, non-zero learning signal:

$$A_i^{(k)} = \frac{c - b_i^{(k)}}{\varepsilon} + \eta \, \Delta b_i^{(k)} = \left( \tfrac{1-\alpha}{\varepsilon} + \eta\alpha \right)\left( c - b_i^{(k-1)} \right),$$

so long as $b_i^{(k-1)} \neq c$. In practice we apply standard clipping on $1/\varepsilon$ (or on $A_i^{(k)}$), to avoid pathological spikes, so the momentum term $\eta \, \Delta b_i^{(k)} = \eta \, \alpha (c - b_i^{(k-1)})$ alone already guarantees a stable non-vanishing signal that pushes $b_i^{(k)}$ toward $c$.

## 3.2 CONFIDENCE-BASED PRIORITIZED EXPERIENCE REPLAY

RLVR faces sample inefficiency challenges—each trajectory is computationally expensive to obtain, yet standard on-policy training uses it only once. Without retaining high-value trajectories, the learning process wastes valuable signals and may forget rare successful strategies. Correct solutions to complex problems are found infrequently, making it beneficial to reinforce them repeatedly rather than rediscover by chance.

We incorporate a *prioritized experience replay* buffer that preferentially retains high-confidence successes. For each prompt $q$, we maintain a small buffer $B[q]$ that stores at most one "best" trajectory —the verified success with highest model confidence.

**Confidence score.** For a response $y = (y_1{:}y_m)$ generated by $\pi_{\theta_k}$, we define

$$C(y|\theta_k) := \frac{1}{m} \sum_{t=1}^{m} \log \pi_{\theta_k}(y_t | y_{<t}, q), \tag{4}$$

A higher $C(y|\theta_k)$ indicates indicates greater model confidence during generation, identifying trajectories the policy can reliably reproduce.

**Buffer management.** When response $y$ achieves $r > 0$ with confidence $C(y|\,\theta_k) \geq C\left(y^\star|\theta_{k^\star}\right)$ (where $y^\star$ is the current buffer entry), we update $\mathcal{B}[q] \leftarrow (y, \theta_k)$. During batch construction, if all $G$ on-policy samples fail for prompt $q$, we replace one failure with the buffered trajectory, maintaining fixed batch size while ensuring at least one positive example per prompt when available.

**Bias control.** Let $\widehat{\nabla}\mathcal{L}_{\text{on}}(\theta)$ denote the on-policy Monte Carlo policy gradient estimator and $\widehat{\nabla}\mathcal{L}_{\text{mix}}(\theta)$ the estimator when replacements occur with probability $\lambda_k \in [0, 1]$. Assuming (i) bounded importance ratios for replaced items and (ii) Lipschitzness of the per-token loss, a standard mixture argument yields

$$\left\| \mathbb{E}\left[ \widehat{\nabla}\mathcal{L}_{\text{mix}}\left(\theta\right) \right] - \mathbb{E}\left[ \widehat{\nabla}\mathcal{L}_{\text{on}}\left(\theta\right) \right] \right\| \leq \lambda_k B \tag{5}$$

for a constant $B$ independent of $k$. As training progresses, $\lambda_k$ decays rapidly since the policy solves more prompts on-policy, yielding controlled vanishing bias with substantial stability gains.

## 3.3 ENTROPY-GUIDED ADVANTAGE WEIGHTING

Standard policy gradients assign uniform advantages across all tokens, which is inefficient for complex reasoning where tokens vary in importance and uncertainty. In multi-step mathematical derivations, models may be confident about routine algebraic steps but uncertain about critical problem-solving decisions. Uniform weighting dilutes learning signals by allocating equal gradient budget to both trivial and crucial steps. We modulate token-level advantages based on model

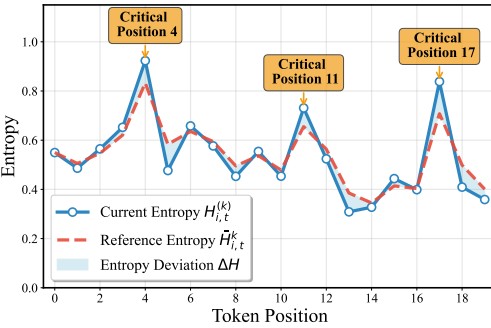 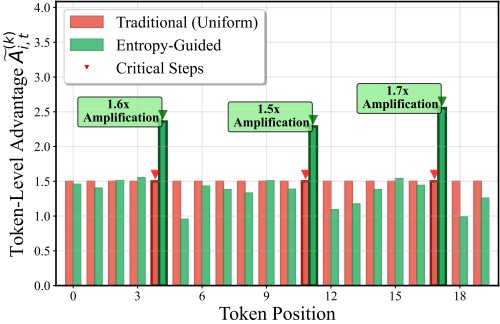

(a) Entropy deviations to weights transformation.  (b) Advantage distribution comparison.

Figure 2: Entropy-guided advantage weighting mechanism. (a) Shows how entropy deviations $\Delta H_{i,t}^{(k)}$ relative to EMA baselines are converted to normalized weights via softmax, with high-uncertainty tokens receiving amplified attention. (b) Demonstrates the resulting advantage redistribution where critical reasoning steps (marked with triangles) receive 1.5-1.7× more learning signal compared to uniform weighting.

uncertainty, using entropy as a proxy. The approach redistributes sequence advantage proportionally to token uncertainty—higher entropy tokens receive larger advantage portions.

For token $t$ in sequence $i$, let $H_{i,t}^{(k)}$ denote the entropy under $\pi_{\theta_k}$. To identify unusually uncertain tokens, we compare against a reference entropy $\bar{H}_{i,t}^{(k)}$ computed from a baseline policy. The entropy delta $\Delta H_{i,t}^{(k)} := H_{i,t}^{(k)} - \bar{H}_{i,t}^{(k)}$ captures deviations from typical uncertainty. Positive deltas indicate higher-than-usual uncertainty, suggesting more critical decision points that warrant increased learning focus. We convert deltas to normalized weights via temperature-scaled softmax

$$w_{i,t}^{(k)} = \frac{\exp\left(\gamma \Delta H_{i,t}^{(k)}\right)}{\sum_{u=1}^{m_i} \exp\left(\gamma \Delta H_{i,u}^{(k)}\right)}, \tag{6}$$

where $\gamma > 0$ controls concentration around high-uncertainty tokens. The entropy-weighted advantage becomes

$$\widetilde{A}_{i,t}^{(k)} = w_{i,t}^{(k)} A_i^{(k)}$$

This reallocates the sequence advantage $A_i^{(k)}$ to emphasize uncertain tokens while preserving total learning signal ($\sum_t w_{i,t} = 1$).Figure 4 illustrates the complete entropy-guided weighting mechanism and its effects on advantage distribution.

**Variance reallocation.** For normalized weights $\{w_{i,t}\}$ with $\sum_t w_{i,t} = 1$, the weighted gradient satisfies

$$\mathrm{Var}\left(\sum_t w_{i,t} g_{i,t}\right) \leq \sum_t w_{i,t} \mathrm{Var}\left(g_{i,t}\right), \quad \sum_t w_{i,t} \mathbb{E}\left[g_{i,t}\right] = \mathbb{E}\left[A_i^{(k)}\right] \cdot s_i,$$

for sequence-level constant $s_i$. Thus entropy weighting redistributes gradient mass without inflating variance or expected magnitude.

### 3.4 UNIFIED TRAINING OBJECTIVE

MAPO keeps components through a token-level clipped surrogate loss with KL regularization. Let $r_{i,t}(\theta) = \pi_\theta\left(y_{i,t}|y_{i,<t}, q\right) / \pi_{\theta_k}\left(y_{i,t}|y_{i,<t}, q\right)$ denote importance ratios. The loss function is

$$\mathcal{L}(\theta) = \mathbb{E}_{q \sim \mathcal{D}, y \sim \pi_{\theta_k}} \left[ \frac{1}{\sum_i m_i} \sum_i \sum_{t=1}^{m_i} \min\left(r_{i,t}\tilde{A}_{i,t}^{(k)}, \mathrm{clip}_{[1-\epsilon, 1+\epsilon]}\left(r_{i,t}\right) \tilde{A}_{i,t}^{(k)}\right) \right] - \beta D_{\mathrm{KL}}\left(\pi_\theta \| \pi_{\mathrm{ref}}\right) \tag{7}$$

where $\mathrm{clip}_{[a,b]}(x) = \min(\max(x, a), b)$ prevents large policy updates. The advantages $\tilde{A}_{i,t}^{(k)}$ computed by equation 2–equation 3 with entropy reweighting equation 6, and batches incorporate confidence-based replay when needed, providing stable learning signals even under challenging reward conditions.

# 4 EXPERIMENTS

We evaluate our method on math–reasoning benchmarks under a unified verifier–reward setting. This section details the training details, main results against strong baselines, ablations isolating each component, and a qualitative case study. All runs use identical data, compute budget, and decoding unless otherwise noted.

## 4.1 SETTINGS

**Models and Datasets.** We conduct experiments with three models: Qwen2.5-Math-1.5B-Instruct, Qwen2.5-Math-7B (math-specialized base)(Yang et al., 2024), and Qwen2.5-14B (general base) (Qwen et al., 2025). Training is performed exclusively on the DAPO-Math-17K corpus (Yu et al., 2025), with all instances formatted via the Qwen-Math prompt template for both training and evaluation.

**Evaluation.** We evaluate our models on four widely-adopted mathematical reasoning benchmarks: MATH500 (Hendrycks et al., 2021), AMC23 (Ouyang et al., 2022), AIME24 (Li et al., 2024), and AIME25 (Balunović et al., 2025). Owing to its larger scale (500 instances), MATH500 is assessed only with **Pass@1**. For the smaller sets AMC23 (40 problems), AIME24 (30 problems), and AIME25 (30 problems), we report both **Pass@1** and **Best** to ensure a thorough evaluation.

- **Pass@1:** Accuracy obtained via greedy decoding; reflects the model's single-shot capability.

- **Best:** The proportion of problems for which at least one solution is correct among 32 independently sampled responses (temperature=1.0, top-$p = 1.0$), characterizing the robustness of the learned policy.

**Training Details.** Training was conducted for 660 steps with a batch size of 256 for response generation and a minibatch size of 64 for parameter update. We used a temperature equal to 1 and top-p equal to 1 to generate G=16 responses for each problem, while the prompt max length was set to 2048 and the response max length was set to 4096, which is much smaller than the maximum length of 20k used in the original DAPO. For the Momentum Group Baseline component, we set the hyper-parameters $\alpha$ and $\eta$ to 0.9 and 0.1 to incorporate historical information.

## 4.2 MAIN RESULTS

We benchmark MAPO against the Base Model, GRPO, and DAPO across four mathematical reasoning datasets: MATH500, AMC23, AIME24, and AIME25, reporting both greedy decoding (Pass@1) and majority-vote accuracy over 32 samples (Best). MAPO integrates momentum-based baselines, confidence-driven replay, and entropy-weighted policy updates to improve the stability and efficiency of large-scale RL training for LLMs (Table 1).

For **Qwen2.5-Math-1.5B-Instruct**, MAPO achieves the highest Average Best accuracy among all methods, while ranking second in Pass@1. The relatively weaker base performance of the smaller model makes correct trajectories rarer during RL training, limiting the potential advantage of MAPO's exploration. Nevertheless, when such correct samples are found, MAPO's replay and entropy-guided weighting can fully exploit them for policy improvement.

On **Qwen2.5-Math-7B**, MAPO surpasses both GRPO and DAPO in Pass@1 and Best. Compared to GRPO, MAPO achieves relative improvements of **13.94%** in Pass@1 and **6.97%** in Best accuracy. Compared to DAPO, MAPO shows relative improvements of **7.21%** in Pass@1 and **10.38%** in Best accuracy. The stronger mathematical reasoning capability of the 7B base policy yields more correct answers during sampling, which benefits MAPO's entropy-guided advantage weighting component by guiding updates toward informative high-confidence trajectories.

For **Qwen2.5-14B**, MAPO delivers higher Best accuracy than DAPO, while slightly lower in Pass@1. We attribute this to DAPO's tendency toward rapid entropy collapse, which can overfit the reward signal and reduce exploration capacity, leading to premature convergence. In contrast, MAPO controls entropy updates to avoid overconvergence, thereby maintaining sampling diversity while achieving strong final performance.

Table 1: Performance comparison across benchmarks. **Pass@1**: accuracy with greedy decoding; **Best**: best accuracy among 32 sampled outputs. Boldface marks the best result per column within each model scale. Best Acc maxima are additionally shaded.

| Model | MATH500 Pass@1/Best | AMC23 Pass@1/Best | AIME24 Pass@1/Best | AIME25 Pass@1/Best | Average Pass@1/Best |
|---|---|---|---|---|---|
| **Qwen2.5-Math-1.5B-Instruct** | | | | | |
| Base | 64.0/70.8 | 47.5/55.0 | 10.0/16.7 | 3.3/20.0 | 31.20/40.62 |
| GRPO | 64.4/74.2 | **57.5**/62.5 | **16.7**/33.3 | **13.3/33.3** | **37.98**/50.83 |
| DAPO | 63.6/76.6 | 52.5/65.0 | 10.0/**40.0** | 10.0/30.0 | 34.03/52.90 |
| **MAPO** | **67.0/80.4** | 55.0/**70.0** | **16.7**/36.7 | 10.0/30.0 | 37.18/**54.28** |
| **Qwen2.5-Math-7B** | | | | | |
| Base | 58.0/67.8 | 52.5/67.5 | 13.3/33.3 | 10.0/26.7 | 33.45/48.83 |
| GRPO | 74.4/93.2 | 62.5/87.5 | 20.0/53.3 | 10.0/**43.3** | 41.73/69.33 |
| DAPO | 75.8/93.8 | 55.0/**95.0** | **33.3**/50.0 | 13.3/30.0 | 44.35/67.20 |
| **MAPO** | **76.0/95.0** | **67.5/95.0** | 30.0/**66.7** | **16.7**/40.0 | **47.55/74.18** |
| **Qwen2.5-14B** | | | | | |
| Base | 60.4/71.2 | 45.0/60.0 | 10.0/13.3 | 6.7/23.3 | 30.53/41.95 |
| GRPO | 75.0/79.4 | 62.5/72.5 | 13.3/26.7 | 13.3/26.7 | 41.03/51.33 |
| DAPO | **79.6**/83.0 | **70.0**/77.5 | **20.0**/30.0 | **16.7**/30.0 | **46.58**/55.13 |
| **MAPO** | 77.8/**86.4** | 65.0/**82.5** | 16.7/**36.7** | 13.3/**36.7** | 43.20/**60.58** |

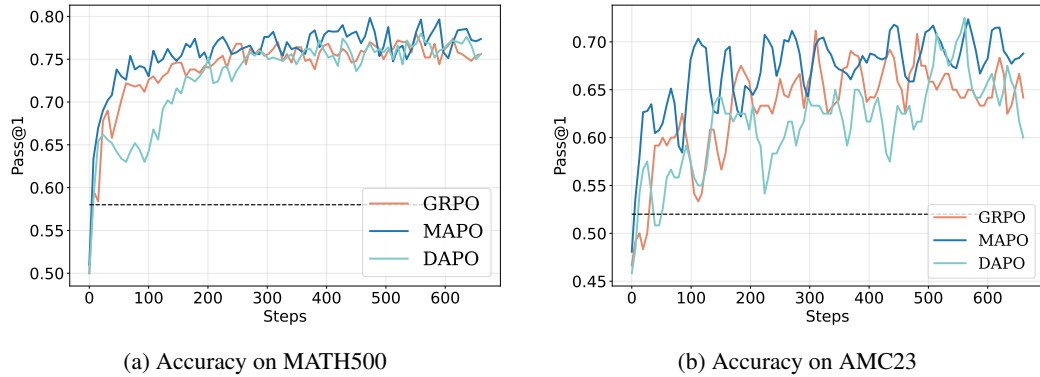

(a) Accuracy on MATH500          (b) Accuracy on AMC23

Figure 3: Comparison of GRPO, DAPO, and our proposed MAPO on MATH500 and AMC23, based on Qwen2.5-Math-7B. MAPO outperforms both GRPO and DAPO in terms of performance at the same number of training steps.

As illustrated in Figure 3, MAPO reaches the inflection point of accuracy substantially earlier than competing methods, reflecting a faster and more stable convergence during RL training. Moreover, continued scaling of the computation, periodic refresh of the query set, and longer generation horizons sustain steady improvement well beyond the early plateau, yielding robust gains without signs of saturation.

### 4.3 IMPACT OF HISTORICAL REWARD INFORMATION

We investigate how historical rewards shape group advantage estimation by comparing three baseline strategies: (i) GRPO using the current-batch mean, (ii) History (Full) using the unwindowed cumulative mean, and (iii) MAPO (EMA) using the momentum-aware EMA baseline from the Method section.

As shown in Fig. 4a, on MATH-500 , MAPO rises more steeply in the early phase and sustains a higher plateau, indicating faster progress under fixed compute. While the full-history variant narrows the gap to GRPO, it consistently trails MAPO, suggesting that recency weighting and the momentum increment—rather than simply aggregating more past data—are responsible for the gains.

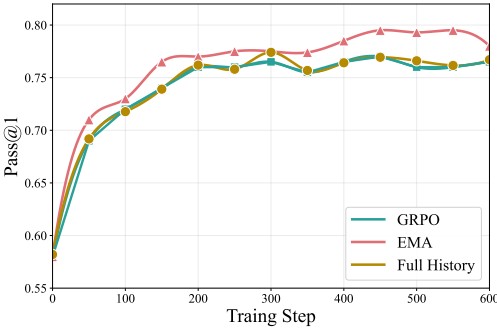
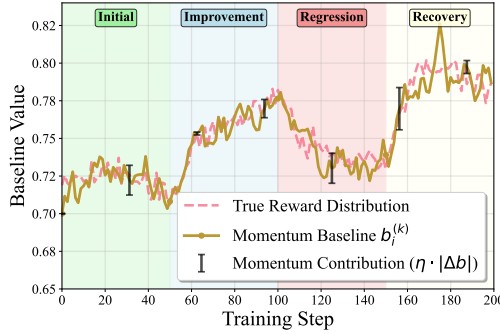

(a) Comparision across different advantage estimation.    (b) Baseline tracking with momentum

Figure 4: Effect of historical information.(a) Pass@1 on MATH-500 comparing GRPO, GRPO with Full History, and Momentum Group Baseline. (b) Momentum Group Baseline tracking during training; shaded regimes mark distribution shifts.

To understand the mechanism, we instrument the EMA baseline over nonstationary training in Fig. 4b. The EMA tracks shifts in the underlying reward trend with small lag across initial → improvement → regression → recovery phases, while the momentum increment $\Delta b$ (red whiskers) provides a persistent learning signal when batch rewards are nearly uniform. This behavior matches our analysis: the EMA term reduces batch noise, and the momentum term mitigates vanishing advantages and expedites recovery after regressions, yielding more stable and sample-efficient policy updates.

## 4.4 ABLATIONS

We conduct a comprehensive ablation study to analyze the contribution of each MAPO component on the Qwen2.5-Math-7B model across three mathematical reasoning benchmarks, AMC23, AIME24, and MATH500. Table 2 presents the results of our systematic component removal analysis. The reinforcement baseline uses GRPO.

**Component Analysis** Adding a Momentum Group Baseline further improves scores on AMC23 (92.50) and especially AIME24 (66.7), suggesting that variance reduction and more stable gradient directions benefit convergence on challenging problems. Introducing Experience Replay maintains high performance on AMC23 (92.50) and slightly boosts MATH500 (93.10), though AIME24 decreases marginally to 63.33. This indicates better reuse of past high-quality solutions can enhance robustness, while distribution shifts in replayed samples may cause small fluctuations for certain datasets. Applying Batch-Entropy Weighting preserves accuracy on AMC23 (92.50) and MATH500 (92.80), but lowers AIME24 to 56.67. This suggests entropy-based weighting can encourage exploration and stabilize easy tasks, yet may require careful tuning for small or high-difficulty benchmarks to avoid overweighting noisy trajectories.

**Full MAPO** The complete MAPO framework, integrating all components, achieves the highest scores: 95 (AMC23), 66.7 (AIME24), and 95.0 (MATH500). Compared to the reinforcement learning baseline, this yields absolute improvements of +7.5, +13.34, and +0.6, respectively. This confirms that MAPO's multi-component design is complementary: momentum baseline stabilizes training, experience replay improves sample efficiency, and entropy weighting guides exploration, jointly yielding robust performance gains across diverse mathematical reasoning tasks.

## 4.5 CASE STUDY

For a correct answer sampled from the trained model, Figure 7 presents the token-level entropy heatmap of Qwen2.5-Math-7B. The figure shows that in its first row, the model exhibits relatively high uncertainty at most token positions. This suggests that in the early stage of reasoning the model remains highly uncertain, hindering its ability to infer a clear problem-solving direction from the preceding context. After the solution approach has been outlined in the preceding context, the model's output entropy drops substantially. However, following a segment of reasoning, the connective words used to bridge into the next reasoning stage exhibit noticeably higher entropy. This indicates that such positions carry a greater training value.

Table 2: Ablation study of MAPO components on Qwen2.5-Math-7B.

| Setting | AMC23 | AIME24 | MATH500 |
|---|---|---|---|
| Initial Policy | 67.5 | 33.3 | 67.8 |
| + Reinforcement (baseline) | 87.5 | 53.3 | 93.2 |
| + Momentum Group Baseline | 92.5 | 66.7 | 92.8 |
| + Experience Replay | 92.5 | 63.3 | 93.2 |
| + Batch-Entropy Weighting | 92.5 | 56.7 | 93.4 |
| **MAPO (Full)** | **95.0** | **66.7** | **95.0** |

After GRPO training (Figure 9), entropy changes concentrate mainly in initial tokens, implying a clearer direction for answer generation and reduced downstream uncertainty. Tokens with higher initial entropy tend to undergo larger adjustments than low-entropy positions. In comparison, Figure 8 illustrates the changes in entropy under the MAPO algorithm with the same data and hyperparameters. Similarly, the entropy of the initial tokens shows substantial variation, but both the magnitude of changes and the number of affected token positions are greater than in GRPO. Additionally, for key tokens such as "Now", "Therefore" and "Since", MAPO produces larger changes. It suggests that MAPO leverages the reward signal more effectively for learning at pivotal points in the reasoning path.

## 5    CONCLUSION

In this work, we present Momentum-Aware Policy Optimization (MAPO), an effective extension of group-based reinforcement learning. MAPO addresses key challenges in policy optimization for reasoning tasks: vanishing advantage estimates under uniform group returns and inefficient credit assignment across long sequences. MAPO incorporates three complementary components: a baseline of the momentum group, confidence-driven replay, and entropy-weighted updates that preserve the simplicity of GRPO while substantially improving the stability of exploration and training.

**Empirical contributions.** The momentum-based baseline prevents gradient collapse when all trajectories succeed or fail uniformly, providing non-vanishing learning signals that accelerate progress in difficult problems. Confidence-weighted replay improves sample efficiency by reusing verified solutions, while entropy-guided weighting focuses updates on uncertain decision steps. Across mathematical reasoning benchmarks, MAPO consistently outperforms GRPO and DAPO in both pass@1 and pass@N metrics, achieving faster convergence with fewer optimization steps.

**Broader implications.** Our results demonstrate that critic-free policy optimization can be significantly enhanced through domain-specific inductive biases without requiring learned value functions. This suggests broader applicability to reinforcement learning settings where binary rewards with low variability hamper standard algorithms. The general principles may be transferred to other domains that require stable training signals, including coding challenges and symbolic reasoning tasks with verifiable feedback.

**Limitations and future work.** The conservative design of the replay buffer, storing one high confidence trajectory per prompt, may restrict the diversity of solutions. Future work could explore multi-trajectory buffers with recency weighting. Moreover, MAPO assumes access to deterministic and verifiable rewards, limiting direct applicability to settings with subjective or continuous feedback. Extending the framework to noisy or learned rewards would require integration with reward models or critics. Finally, stable training still relies on KL-constrained updates, necessitating careful tuning of the trust-region radius.

In summary, MAPO provides a robust, drop-in solution for scaling reinforcement learning to complex, multi-step reasoning with large language models. By ensuring persistent learning signals and intelligent gradient allocation, it addresses core pathologies in group-based policy optimization and paves the way for more reliable training from verifiable feedback.

## ETHICS STATEMENT

This work adheres fully to the ICLR Code of Ethics. Our study does not involve human subjects, personally identifiable data, or sensitive demographic information. All datasets used are publicly available and widely adopted in the literature, with preprocessing steps documented to ensure transparency. The proposed methods aim to improve training stability and efficiency for large language models in mathematical reasoning, without introducing applications that pose foreseeable risks of misuse or societal harm. We disclose no conflicts of interest or external sponsorship that could influence this work.

## REPRODUCIBILITY

We have taken concrete steps to ensure reproducibility. The main paper clearly describes the model architecture, optimization objectives, and evaluation protocols. Complete proofs and derivations of theoretical results are provided in the appendix. Experimental details, including hyperparameters, training schedules, and evaluation setups, are reported in our paper. All datasets used are publicly available. Together, these resources enable independent verification of our theoretical and empirical results.

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

# A NOTATION

Table3 summarizes the notation used throughout the paper. We use $i$ to index responses in a group, $t$ to index token positions, and $k$ to index training iterations.

Table 3: Notation.

| Symbol | Description |
|---|---|
| **Data & Policies** | |
| $x \sim D$ | Prompt from distribution $D$ |
| $y_i$ | Generated sequence |
| $s_t = (x, y_{<t}), a_t = y_t$ | State/action at step $t$ |
| $\pi_\theta, \pi_{\text{old}}, \pi_{\text{ref}}$ | Trainable, old, reference policies |
| **Rewards & Advantages** | |
| $G$ | Group size |
| $r_i$ | Reward for $y_i$ |
| $\mu, \sigma$ | Mean/std of group rewards |
| $\mu_{-i}$ | LOO mean (excl. $r_i$) |
| $\hat{A}_t$ | Token advantage (GAE/group-normalized) |
| $V_\phi$ | Critic baseline (PPO) |
| $b_i^{(k)}$ | Momentum Group Baseline |
| $\Delta b_i^{(k)}$ | Baseline increment |
| $A_i^{(k)}$ | Momentum advantage |
| $\alpha, \eta, \epsilon$ | EMA decay, momentum coeff, stabilizer |
| **Entropy Weighting** | |
| $H_{i,t}^{(k)}$ | Token entropy at $t$ |
| $\bar{H}_{i,t}^{(k)}$ | EMA entropy |
| $\Delta H_{i,t}^{(k)}$ | Entropy delta |
| $w_{i,t}^{(k)}$ | Token weight (softmax of $\gamma \Delta H$) |
| $\tilde{A}_{i,t}^{(k)}$ | Weighted advantage |
| $\beta, \gamma$ | EMA decay, softmax temp |
| **Replay & Optimization** | |
| $B$ | Replay buffer (prompt $\rightarrow$ best trajectory) |
| $C(y|\theta)$ | Confidence score |
| $\text{clip}_{[1-\epsilon,1+\epsilon]}$ | Clipping operator (bounds $1 \pm \epsilon$) |
| $\beta_{\text{KL}}$ | KL penalty |
| $L(\theta)$ | Loss Function |

# B FINITE-GROUP EFFECTS OF GROUP-NORMALIZED ADVANTAGES

We analyze how the finite group size in GRPO affects advantage estimates, to motivate the momentum baseline and replay components. Consider a fixed prompt where each of the $G$ sampled trajectories receives an i.i.d. binary reward $r_i \sim \text{Bernoulli}(p)$, $i = 1, \ldots, G$, with true success probability $p \in (0, 1)$. we compute the group mean $\mu = \frac{1}{G} \sum_i r_i$ and standard deviation $\sigma = \sqrt{\mu(1-\mu)}$, and assign each trajectory a normalized advantage

$$A_i = \frac{r_i - \mu}{\sigma + \varepsilon}.$$

where $\epsilon > 0$ is a small stabilizer. This group-normalized advantage is the same for every token in trajectory $i$ (treating the entire sequence as one decision). In the limit of an infinite group ($G \rightarrow \infty$), $\mu \rightarrow p$ and $\sigma \rightarrow \sqrt{p(1-p)}$, so one can define the population advantage values for a correct (reward 1) or incorrect (reward 0) trajectory:

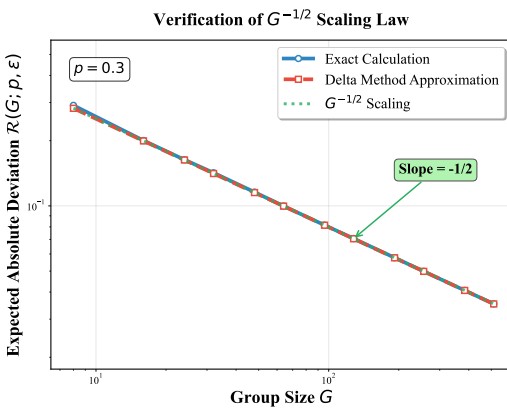

(a) **Scaling law.** Exact computation and delta–method approximation collapse onto a slope $-\frac{1}{2}$ line, confirming $\mathbb{E}[\Delta(G, k; p)] = \Theta(G^{-1/2})$ across representative $p$.

(b) **Finite–group distortion.** For $G=32, p=0.4$, finite-sample advantages for correct/incorrect responses deviate from population values; at homogeneous groups $(k=0, G)$ the normalized advantage vanishes $(\sigma=0)$, illustrating exact gradient collapse.

Figure 5: **Finite-group effects in group-normalized advantages.** (a) Empirical verification of the $G^{-1/2}$ deviation predicted by the delta method; (b) qualitative shape of finite-sample advantages and the collapse at extreme $k$. Together, these support our design choices: momentum baselines to supply a non-degenerate signal near $\mu \approx 0, 1$, and confidence-aware replay to preserve rare successes when an on-policy group fails.

$$A_{\text{right}}(p) = \frac{1 - p}{\sqrt{p(1 - p) + \varepsilon}}, \quad A_{\text{wrong}}(p) = \frac{-p}{\sqrt{p(1 - p) + \varepsilon}}.$$

These are the ideal normalized advantages as $G \to \infty$. For any finite group of size $G$, suppose exactly $k$ out of $G$ trajectories are successful, therefore, $\mu = \frac{k}{G}$. The normalized advantage for a trajectory in this finite sample will deviate from the population value. In fact, for a randomly chosen token in the batch, the expected absolute deviation in advantage is

$$\Delta(G, k; p) = \frac{k}{G} \left| A_{\text{right}}(k/G) - A_{\text{right}}(p) \right| + \frac{G - k}{G} \left| A_{\text{wrong}}(k/G) - A_{\text{wrong}}(p) \right|.$$

This quantity $\Delta(G, k; p)$ measures how much the per-token learning signal is distorted due to sampling variance in a group of finite size. Averaging over $k \sim \text{Binomial}(G, p)$ yields the exact expectation

$$\mathcal{R}(G; p, \varepsilon) = \mathbb{E}\big[\Delta(G, k; p)\big] = \sum_{k=0}^{G} \binom{G}{k} p^k (1 - p)^{G-k} \, \Delta(G, k; p).$$

Using a second-order delta-method approximation with $\text{Var}(\mu) = p(1 - p)/G$ one obtains

$$\mathcal{R}(G; p, \varepsilon) = \sqrt{\frac{2}{\pi}} \sqrt{\frac{p(1 - p)}{G}} \left( p \left| A'_{\text{right}}(p) \right| + (1 - p) \left| A'_{\text{wrong}}(p) \right| \right) + O(G^{-1}),$$

implying that **the expected advantage deviation scales on the order of** $G^{-1/2}$; the signed bias is $O(G^{-1})$. Moreover, the probability that the group is homogeneous (all correct or all incorrect) is

$$\Pr[\text{homogeneous}] = p^G + (1 - p)^G,$$

on which $\sigma = 0$ and $\hat{A}_i = 0$ for all responses, causing exact gradient collapse. Both effects justify our momentum baseline (non-degenerate signal when $\mu \approx 0$ or 1) and confidence-aware replay (retain rare successes when an on-policy group fails). Empirical evaluations in Fig. 5a–5b match the $G^{-1/2}$ law and show exponentially vanishing collapse probability away from $p \in \{0, 1\}$.

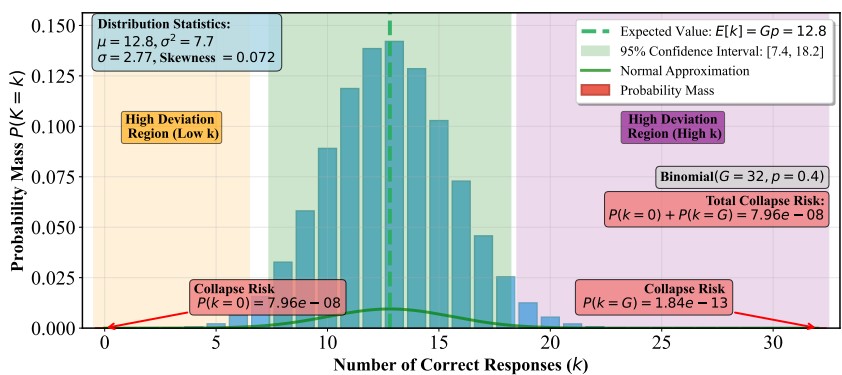

(a) Delta–method relative error vs. $G$. Error $< 5\%$ for $G \geq 16$.

(b) Binomial mass illustrating extreme-$k$ regions where gradient collapse occurs.

Figure 6: (a) Accuracy of the delta–method approximation used to derive the $G^{-1/2}$ scaling; (b) probability mass near homogeneous groups explaining collapse at $\mu \approx 0, 1$.

## C  THEORETICAL ANALYSIS

We provide theoretical guarantees for MAPO's training objective and further analysis of its components. In particular, we show that under standard smoothness and trust-region conditions, MAPO's surrogate loss is monotonically non-decreasing, akin to policy gradient methods with trust regions. We also analyze how the momentum baseline acts as a variance-reducing control variate, how the replay buffer introduces only a small and vanishing bias, and how entropy-based token weighting preserves the scale of gradient updates while improving credit assignment.

**Theorem 1** (Monotone surrogate improvement under trust region). *Suppose (i) the surrogate objective $\mathcal{L}$ in equation 7 is $L$-smooth in $\theta$, (ii) importance ratios are clipped as in equation 7 with bounds ensuring $D_{\mathrm{KL}}(\pi_{\theta_k} \,\|\, \pi_\theta) \leq \delta$, and (iii) $\eta, \alpha, \beta, \gamma$ are bounded with $\sigma^{(k)} \geq \sigma_{\min} > 0$. Then for a sufficiently small step size $\eta_k$, the update $\theta_{k+1} = \theta_k + \eta_k \widehat{\nabla}\mathcal{L}(\theta_k)$ satisfies*

$$\mathbb{E}\big[\mathcal{L}(\theta_{k+1}) - \mathcal{L}(\theta_k)\big] \geq \eta_k \,\mathbb{E}\big[\|\nabla\mathcal{L}(\theta_k)\|_2^2\big] - \tfrac{L}{2}\eta_k^2 \,\mathbb{E}\big[\|\widehat{\nabla}\mathcal{L}(\theta_k)\|_2^2\big] - \lambda_k B,$$

*where $\lambda_k$ denotes the fraction of off-policy (replay) data at iteration $k$, and $B$ is a constant bounding the replay-induced bias (cf. equation 5). Consequently, for sufficiently small $\eta_k$ and as $\lambda_k \to 0$, the expected surrogate objective improves, i.e., $\mathbb{E}\big[\mathcal{L}(\theta_{k+1}) - \mathcal{L}(\theta_k)\big] \geq 0$.*

**Proposition 1** (Response utilization and nonzero signal). *Let $\mathrm{RUR}_k$ be the fraction of responses in a batch with non-zero $\widetilde{A}_{i,t}^{(k)}$. If (a) $b^{(k)}$ uses EMA equation 2 and (b) replay injects a buffer trajectory whenever an all-failure group appears, then $\mathrm{RUR}_k \geq 1 - p_0^G$, where $p_0$ is the failure rate per sample under $\pi_{\theta_k}$. Thus $\mathrm{RUR}_k$ remains bounded away from zero even when $p_0$ is high.*

**Remark.** In expectation, $1 - p_0^G$ is the probability that at least one out of $G$ i.i.d. trajectories is successful. The replay injection rule (b) guarantees that every prompt yields at least one successful trajectory in the learning batch (unless none exist even in the buffer). Thus the algorithm effectively behaves as if each prompt has success probability $1 - p_0^G$ of providing a learning signal. Even when $p_0$ is close to 1, the probability of all-$G$ failures is $p_0^G$; MAPO substitutes a success from memory when available, ensuring a non-zero advantage in that group.

## C.1 EMA CONVERGENCE AND CONTROL VARIATE VIEW

The momentum baseline in MAPO introduces a bias–variance trade-off that reduces gradient variance. In a stationary regime with $p_k \equiv p$, the EMA baseline satisfies $b^{(k)} = \alpha \mu^{(k)} + (1 - \alpha) b^{(k-1)}$, hence

$$\mathbb{E}[b^{(k)}] \to p, \qquad \big| \mathbb{E}[b^{(k)}] - p \big| \le (1 - \alpha)^k \big| \mu^{(0)} - p \big|,$$

so the baseline bias decays geometrically at rate $(1 - \alpha)$. For trajectory $i$,

$$\frac{r_i^{(k)} - b_i^{(k)}}{\sigma^{(k)}} = \underbrace{\frac{r_i^{(k)} - \mu_{-i}^{(k)}}{\sigma^{(k)}}}_{Z_i^{(k)}} + \underbrace{\frac{b_i^{(k)} - \mu_{-i}^{(k)}}{\sigma^{(k)}}}_{U_i^{(k)}}.$$

With $\mathbb{E}[Z_i^{(k)}] = 0$ and $\mathbb{E}[U_i^{(k)}] \to 0$, the EMA adds a decaying bias while reducing variance through $\mathrm{Cov}(Z_i^{(k)}, U_i^{(k)}) < 0$ (empirically typical). The additive momentum term $\eta \Delta b_i^{(k)}$ further acts as a control variate; variance is minimized for $\eta$ near $-\mathrm{Cov}(Z_i^{(k)}, \Delta b_i^{(k)})/\mathrm{Var}(\Delta b_i^{(k)})$.

## C.2 KL-REGULARIZED OFF-POLICY MIXING

Consider a trajectory $(x, \tilde{y})$ from the replay buffer $\mathcal{B}$, generated by $\pi_{\theta_{\mathrm{gen}}}$. Its importance ratio under the current policy $\pi_\theta$ is

$$w = \frac{\pi_\theta(\tilde{y} \mid x)}{\pi_{\theta_{\mathrm{gen}}}(\tilde{y} \mid x)} \quad \text{with} \quad \log w = \sum_{t=1}^{T} \Delta_t,$$

where $\Delta_t := \log \pi_\theta(\tilde{y}_t \mid \tilde{y}_{<t}, x) - \log \pi_{\theta_{\mathrm{gen}}}(\tilde{y}_t \mid \tilde{y}_{<t}, x)$.

We enforce a tokenwise KL trust region via the chain rule

$$\sum_{t=1}^{T} \mathbb{E}_{\pi_{\theta_{\mathrm{gen}}}} \big[ D_{\mathrm{KL}}\big(\pi_\theta(\cdot \mid \tilde{y}_{<t}, x) \, \big\| \, \pi_{\theta_{\mathrm{gen}}}(\cdot \mid \tilde{y}_{<t}, x)\big) \big] \le \delta.$$

Assume

**(A1) Local quadratic KL.** There exists $L_{\mathrm{KL}} > 0$ such that, for all relevant contexts $s$,

$$D_{\mathrm{KL}}\big(\pi_\theta(\cdot \mid s) \, \big\| \, \pi_{\theta_{\mathrm{gen}}}(\cdot \mid s)\big) \le \frac{L_{\mathrm{KL}}}{2} \|\theta - \theta_{\mathrm{gen}}\|_2^2.$$

**(A2) Log-probability Lipschitzness.** For all $\theta, \theta', y, s$,

$$\big| \log \pi_\theta(y \mid s) - \log \pi_{\theta'}(y \mid s) \big| \le L_{\log} \|\theta - \theta'\|_2.$$

Then from (A1) and the trust region, $\|\theta - \theta_{\mathrm{gen}}\|_2 \le C_\theta \sqrt{\delta/T}$ for some constant $C_\theta > 0$, and by (A2) the per-token log-ratio is bounded

$$|\Delta_t| \le b, \qquad b := L_{\log} C_\theta \sqrt{\frac{\delta}{T}}.$$

Define the centered sum

$$Z_T := \sum_{t=1}^{T} \Big( \Delta_t - \mathbb{E}_{\pi_{\theta_{\mathrm{gen}}}}[\Delta_t] \Big).$$

Since $|\Delta_t - \mathbb{E}_{\pi_{\theta_{\mathrm{gen}}}}[\Delta_t]| \le 2b$, Azuma–Hoeffding implies that, for any $\alpha \in (0, 1)$,

$$\mathbb{P}\Big( |Z_T| \ge 2b \sqrt{2T \log(2/\alpha)} \Big) \le \alpha.$$

Because $\mathbb{E}_{\pi_{\theta_{\text{gen}}}}[\log w] = -D_{\text{KL}}(\pi_{\theta_{\text{gen}}} \| \pi_\theta)$, we obtain, with probability at least $1 - \alpha$,

$$\log w \in \left[ -D_{\text{KL}}(\pi_{\theta_{\text{gen}}} \| \pi_\theta) \pm C \sqrt{\delta \log(2/\alpha)} \right], \qquad C := 2\sqrt{2} L_{\log} C_\theta.$$

Hence $w$ concentrates near 1 with $|\log w| = O(\sqrt{\delta})$ (high probability). For small $\delta$, a Taylor expansion of $e^x$ yields $\mathbb{E}[(w-1)^2] = O(\delta)$ under bounded log-ratio increments.

### C.3 SCALE PRESERVATION IN ENTROPY-WEIGHTED TOKEN UPDATES

The per-token policy-gradient loss is $\ell_t = -\log \pi_\theta(y_t \mid y_{<t}, x) A_t$. MAPO reweights it as

$$\tilde{\ell}_t = w_t \ell_t, \qquad w_t = \frac{\exp(\gamma \Delta H_t)}{\sum_u \exp(\gamma \Delta H_u)}, \qquad \sum_t w_t = 1.$$

The reweighted sequence loss is a convex combination:

$$\tilde{\mathcal{L}} = \sum_t w_t \ell_t \quad \Rightarrow \quad \min_t \ell_t \leq \tilde{\mathcal{L}} \leq \max_t \ell_t.$$

Let $g_t = \nabla_\theta \ell_t$ and $g = \sum_t w_t g_t$. Then

$$\mathbb{E}\|g\|^2 = \sum_{t,u} w_t w_u \mathbb{E}[g_t^\top g_u] \leq \sum_{t,u} w_t w_u \sqrt{\mathbb{E}\|g_t\|^2} \sqrt{\mathbb{E}\|g_u\|^2} = \left( \sum_t w_t \sqrt{\mathbb{E}\|g_t\|^2} \right)^2 \leq \sum_t w_t \mathbb{E}\|g_t\|^2,$$

where we used Cauchy–Schwarz and Jensen. In particular, if $\mathbb{E}\|g_t\|^2 \leq \sigma_g^2$ for all $t$, then $\mathbb{E}\|g\|^2 \leq \sigma_g^2$. Hence entropy-based weighting with $\sum_t w_t = 1$ does not inflate the gradient scale; it redistributes signal across tokens according to uncertainty.

## D ALGORITHM

1 summarizes the training progress of our method.

Algorithm1 provides an overview of MAPO's training procedure, integrating the momentum baseline, prioritized replay, and entropy weighting into the standard policy optimization loop. For clarity, we present the update in two conceptual stages:

- Step A computes the momentum advantage and performs a preliminary policy update using uniform token weights
- Step B then computes entropy-based weights and applies the reweighted objective for a final update. In implementation, these can be combined or repeated as needed, but separating them highlights that the entropy weighting can be viewed as an enhancement on top of a baseline RL update.

## E CASE STUDY

For a correct answer sampled from the model, we examine the entropy changes on three different models: Baseline, GRPO, and MAPO. Figure 7 shows the entropy values of the Baseline model for this answer. Figure 9 presents the magnitude of entropy changes after GRPO training. In comparison, Figure 8 contains more dark-colored points, indicating larger entropy changes, and shows greater variation at key tokens, such as "Now", "Since" and "Therefore".

**Algorithm 1** MAPO: Momentum-Aware Advantage Policy Optimization

**Require:** Policy $\pi_\theta$, reference $\pi_{\text{ref}}$; group size $G$; clip $\epsilon$; KL weight $\beta$; EMA $\alpha \in (0,1]$; momentum $\eta \geq 0$; stabilizer $\varepsilon > 0$; entropy temp $\tau > 0$.

1: **Init:** $\pi_{\text{old}} \leftarrow \pi_\theta$; $\mathcal{B}_{\text{base}} \leftarrow \emptyset$; $\mathcal{M} \leftarrow \emptyset$.

2: **while** training **do**

3:     **Rollout:** For each prompt $x$, sample $\{y_i\}_{i=1}^G \sim \pi_{\text{old}}(\cdot|x)$, get rewards $r_i$.

4:     **Stats:** $\mu = \frac{1}{G} \sum_i r_i$, $\mu_{-i} = \frac{1}{G-1} \sum_{j \neq i} r_j$, $\sigma^2 = \frac{1}{G} \sum_i (r_i - \mu)^2$.

5:     **EMA baseline:** $b^{\text{prev}} \leftarrow \mathcal{B}_{\text{base}}[x]$ (default $= \mu$);

6:               $b_i \leftarrow \alpha \mu_{-i} + (1-\alpha) b^{\text{prev}}$;

7:               $\Delta b_i \leftarrow b_i - b^{\text{prev}}$;

8:               $\mathcal{B}_{\text{base}}[x] \leftarrow \frac{1}{G} \sum_i b_i$.

9:     **Advantages:** $\widehat{A}_i \leftarrow \dfrac{r_i - b_i}{\sigma + \varepsilon} + \eta \cdot \Delta b_i$   (optionally center).

10:     **Replay:** If $\forall i$, $r_i = 0$ and $(x, \tilde{y}) \in \mathcal{M}$, replace one $(x, y_i, r_i)$ with $(x, \tilde{y}, 1)$.

11:     **Step A:** $s_i(\theta) = \exp\left\{ \frac{1}{|y_i|} \sum_t \log \frac{\pi_\theta(y_{i,t}|\cdot)}{\pi_{\text{old}}(y_{i,t}|\cdot)} \right\}$;

12:         $\ell_i^{(A)} = \min\left( s_i \widehat{A}_i, \text{clip}(s_i, 1-\epsilon, 1+\epsilon) \cdot \widehat{A}_i \right)$;

13:         $\theta' \leftarrow \arg\max_\theta \left\{ \frac{1}{G} \sum_i \ell_i^{(A)} - \beta \cdot D_{\text{KL}}(\pi_\theta \| \pi_{\text{ref}}) \right\}$.

14:     **Step B:** Compute $\Delta H_{i,t}$; $w_{i,t} \leftarrow \text{softmax}_t(\Delta H_{i,t}/\tau)$;

15:         $\rho_{i,t}(\theta') = \frac{\pi_{\theta'}(y_{i,t}|\cdot)}{\pi_{\text{old}}(y_{i,t}|\cdot)}$;

16:         $\tilde{A}_{i,t} = w_{i,t} \cdot \widehat{A}_i$;

17:         $\ell_{i,t}^{(B)} = \min\left( \rho_{i,t} \tilde{A}_{i,t}, \text{clip}(\rho_{i,t}, 1-\epsilon, 1+\epsilon) \cdot \tilde{A}_{i,t} \right)$;

18:         $\theta \leftarrow \arg\max_{\theta'} \left\{ \frac{1}{G} \sum_i \frac{1}{|y_i|} \sum_t \ell_{i,t}^{(B)} - \beta \cdot D_{\text{KL}}(\pi_{\theta'} \| \pi_{\text{ref}}) \right\}$.

19:     **Update:** Periodically $\pi_{\text{old}} \leftarrow \pi_\theta$; for success $(x, y_i, 1)$,

20:         store $C(x, y_i) = \frac{1}{|y_i|} \sum_t \log \pi_{\text{old}}(y_{i,t}|\cdot)$,

21:         update $\mathcal{M}[x]$ if $C$ improves.

22: **end while**

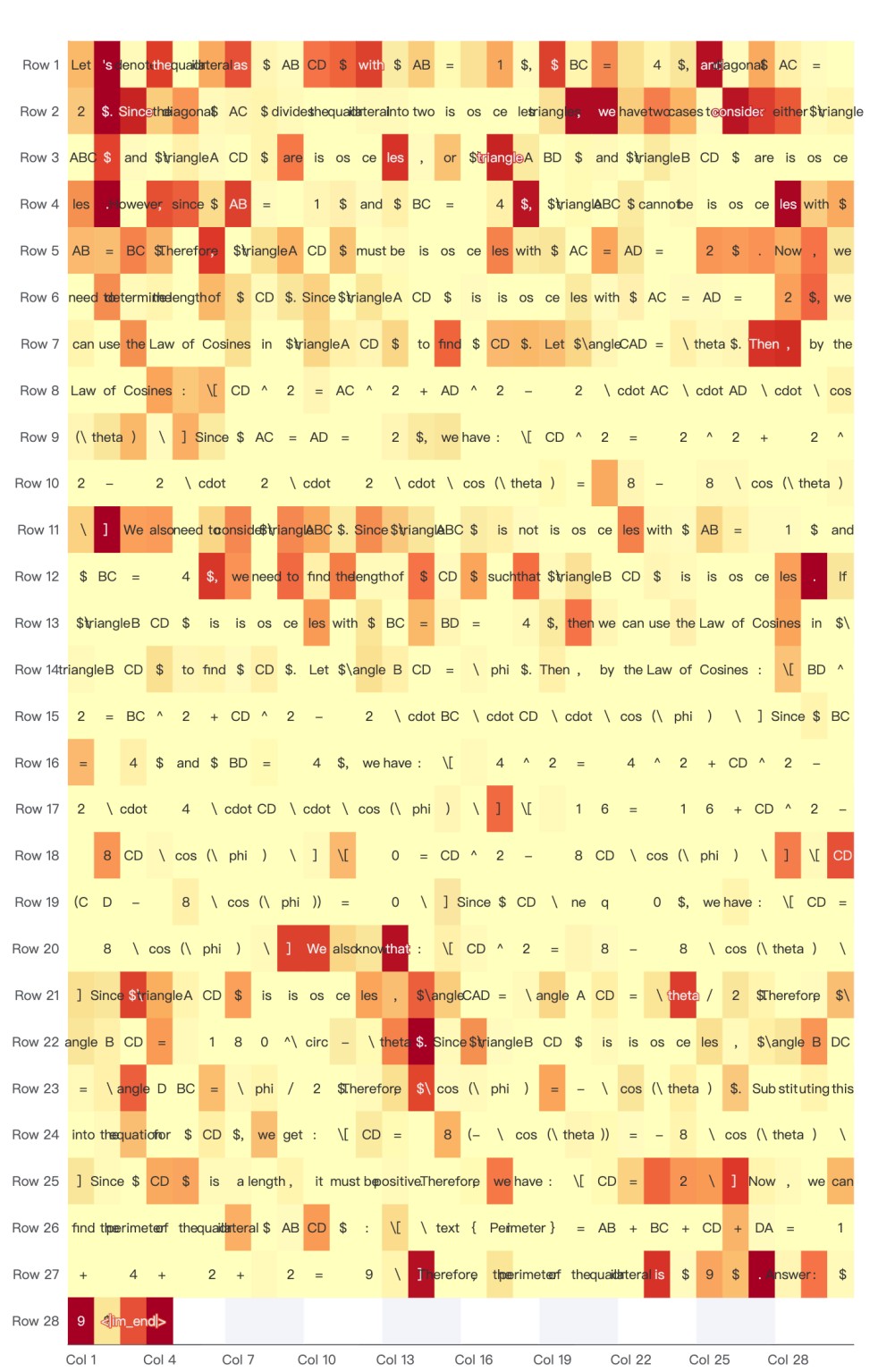

Figure 7: Token-level entropy heatmap of a correct answer on Qwen2.5-Math-7B

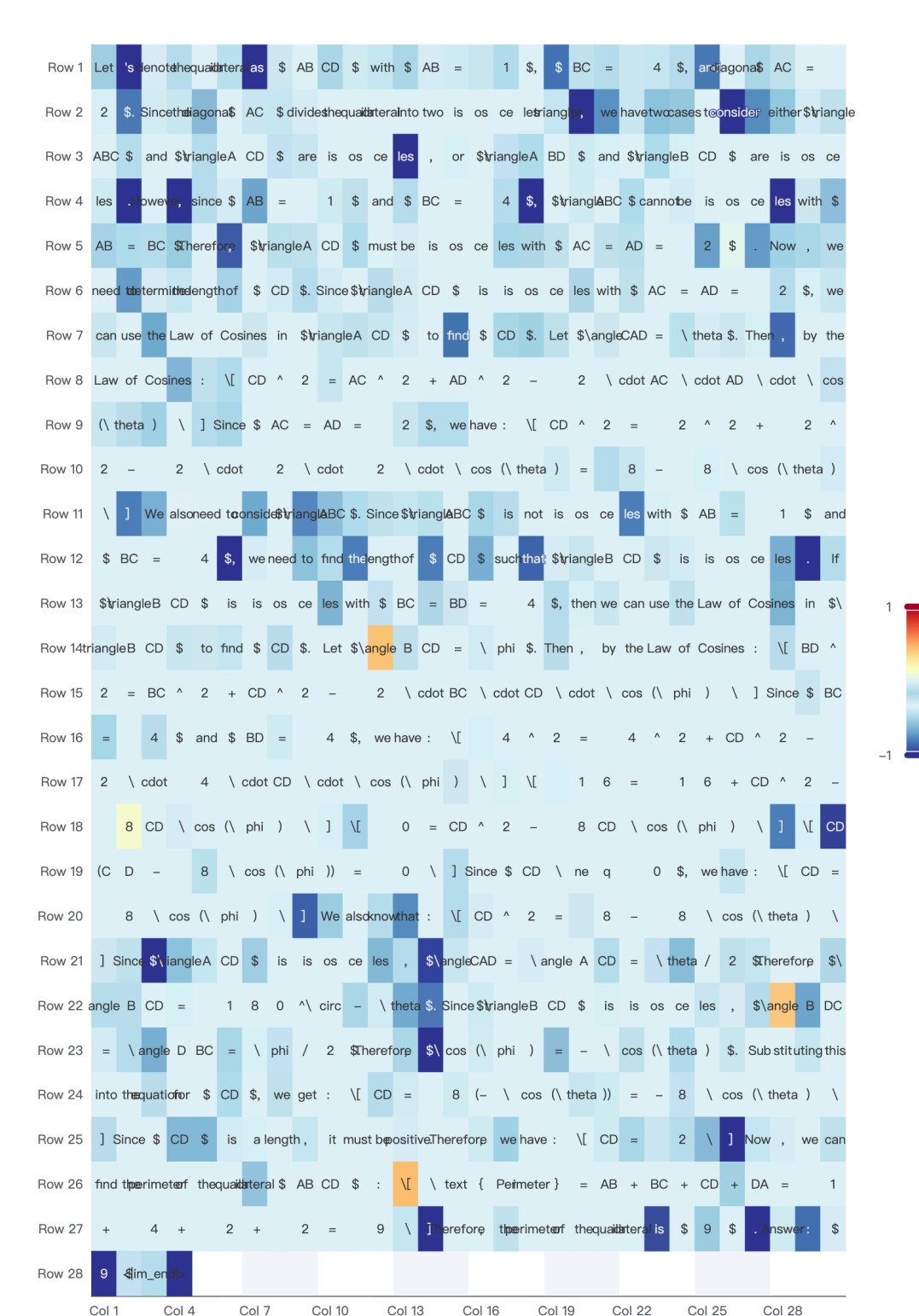

Figure 8: Entropy change heatmap at each token position for a correct answer before and after **MAPO** training of Qwen2.5-Math-7B

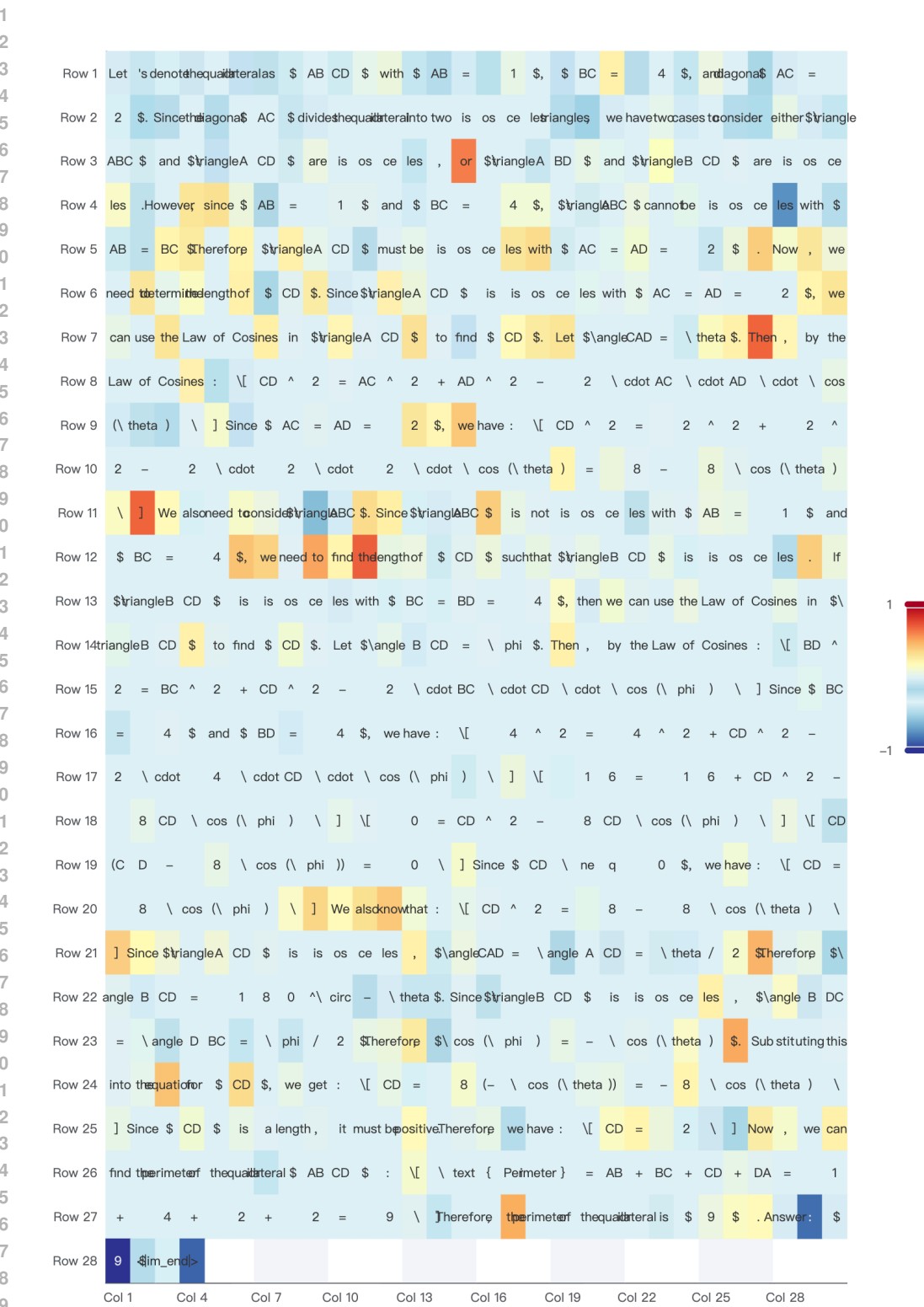

Figure 9: Entropy change heatmap at each token position for a correct answer before and after **GRPO** training of Qwen2.5-Math-7B

