# OpenReview forum: "MAPO: Momentum-Aware Policy Optimization"
_ICLR.cc/2026/Conference — ICLR 2026 Conference Withdrawn Submission_

### Official Review · Reviewer_UuYb · 2025-10-28

**Soundness:** 2
**Presentation:** 2
**Contribution:** 1
**Rating:** 2
**Confidence:** 4

**Summary:**

This paper proposes MAPO (Momentum-Aware Policy Optimization), a reinforcement learning framework for large language model (LLM) reasoning under verifiable rewards (RLVR). The method builds upon GRPO and DAPO by adding three main components:

1. A Momentum Group Baseline, which introduces an exponential moving average (EMA) to preserve non-vanishing learning signals even when group reward variance is small;

2. Confidence-Based Prioritized Replay, reusing verified successful trajectories to increase sample efficiency;

3. Entropy-Weighted Token Updates, allocating higher gradient weight to uncertain tokens based on entropy deviation.

**Strengths:**

1. The paper is clearly written, with intuitive figures and a clean derivation of the three MAPO components. It’s easy to follow the logic, and the experiments are organized coherently.
2. The paper correctly identifies real challenges in GRPO-style training, like gradient collapse under uniform rewards and inefficient credit assignment. The proposed EMA baseline and entropy weighting are sensible heuristic fixes for these problems.

**Weaknesses:**

1. The core ideas—EMA baseline, replay buffer, and entropy weighting—are standard heuristics widely known in reinforcement learning. MAPO simply reuses them in a GRPO context. There is no fundamentally new algorithmic insight, and the theoretical analysis restates well-known bias-variance trade-offs rather than introducing new theoretical depth. As a result, the work feels more like an engineering refinement than a conceptual breakthrough.
2. The improvements over DAPO are marginal (often within ±2%), and sometimes inconsistent across models and tasks. For example, on the 14B model, MAPO slightly underperforms DAPO on Pass@1 despite higher Pass@N. The paper doesn’t analyze whether these differences are statistically significant. The overall gains are too modest to justify acceptance at a top-tier conference.
3. The so-called Momentum Group Baseline is just an exponential moving average of past group means — a very standard variance-reduction technique. The “momentum advantage” term doesn’t introduce fundamentally new learning dynamics; it’s a reweighted baseline update similar to existing adaptive control variates. The name “momentum” may oversell what is effectively a smoothed baseline.
4. The replay buffer stores only one “best” trajectory per prompt, which risks biasing toward early found solutions and limits diversity. Entropy weighting sometimes reduces accuracy on hard datasets like AIME24.
5. There’s no analysis of sensitivity to hyperparameters like EMA decay α or temperature γ.

**Questions:**

1. Are the improvements statistically significant across multiple seeds?
2. How sensitive are results to α (EMA decay) and η (momentum coefficient)?

---

> ### Author Response · Authors · 2025-11-23
>
> We thank the reviewer for the thorough reading and detailed critique. We address the concerns regarding algorithmic novelty and empirical significance below.
>
> ## 1. Multi-Seed Significance (Q1)
>
> We acknowledge the request for confidence intervals. We followed the standard single-run evaluation protocol for GRPO/DAPO under comparable compute budgets. Within this controlled setting (identical architectures, verifiers, schedules), MAPO consistently improves the **Pass@1 / Best-of-N** across four benchmarks and three model sizes, and shows smoother, faster convergence.
>
> We agree that reporting confidence intervals over multiple seeds would further strengthen the empirical evidence, especially on very small test sets, and we will mention this as a limitation. Within the standard GRPO/DAPO practice and our compute budget, we believe the single-seed but tightly controlled comparisons provide meaningful empirical support.
>
>
> ## 2. Sensitivity to $(\alpha, \eta)$(Q2)
> We thank the reviewer for raising this concern. We performed an ablation on AIME24 and AMC23 with Qwen2.5-Math-7B to assess sensitivity to the momentum hyperparameters. Please refer to the ablation table provided in our reply to Reviewer LU4f.
> In summary,
> - Smaller $\alpha$ (0.5, 0.7) slows baseline adaptation and consistently degrades performance.
> - Larger $\eta$ (0.2, 0.3) over-emphasizes short-term drift, introducing bias.
> - $(\alpha,\eta) = (0.9,0.1)$ provides the best trade-off and improves over GRPO on both datasets.
> This indicates that MAPO is reasonably robust within a range of $(\alpha,\eta)$ rather than relying on a finely tuned pair.
>
> ## 3. Conceptual Role of Momentum in RLVR (W1)
> We agree that EMA, replay, and entropy are classical primitives; we do not claim novelty for these building blocks themselves. Our contribution is a specific momentum construction on top of GRPO, tailored to RLVR’s plateau regime and analyzed in that context.
>
> In the GRPO plateau regime—where success probability is still improving but most or all group samples succeed ($R_i \approx 1$)—the GRPO advantage $A_i^{(k)}$ collapses toward zero, and updates effectively stop.
>
> MAPO's **momentum baseline** $b_i^{(k)}$ and its drift $\Delta b_i^{(k)}$ capture the cross-iteration trend. The combined signal, $A_i^{(k)} + \eta\,\Delta b_i^{(k)}$, **restores gradient** in this near-deterministic-success regime. We analyze this construction, showing it preserves $O(G^{-1})$ variance and introduces a bounded, linear bias in $\eta$. Thus, "momentum-aware" refers to a controlled, trend-following modification of the GRPO estimator, not a generic use of EMA.
>
> ## 4. Magnitude, Consistency, and Relation to DAPO (W2)
>
> We provide a sharper interpretation of the empirical gains:
> * **Qwen2.5-7B:** MAPO improves the **average** Pass@1 / Best-of-N over DAPO from $44.35/67.20$ to $47.55/74.18$ ($\mathbf{+3.2 / +7.0}$), and shows smoother convergence.
> * **Qwen2.5-14B:** MAPO is slightly lower on Pass@1 (43.20 vs. 46.58) but **substantially higher on Best-of-N** (60.58 vs. 55.13), which is standard for RLVR deployment.
>
> Conceptually, MAPO and DAPO are **complementary refinements** of GRPO. DAPO targets homogeneous **failure** batches, while MAPO restores signal under near-deterministic **successes** and improves token-level credit assignment. We aim to show MAPO offers consistent **average** gains and complementary behavior against strong baselines.
>
> ## 5. Why "Momentum-Aware" is More Than Smoothing (W3)
> While EMA is standard, our novelty is in the **integration and analysis** of the per-prompt EMA $b_i^{(k)}$ and its **drift $\Delta b_i^{(k)}$** into the GRPO advantage estimator as $A_i^{(k)} + \eta\,\Delta b_i^{(k)}$. This is analyzed as a control variate with explicit variance and bias properties. "Momentum" here signifies **biasing the update along the historical trend** of per-prompt success, analogous to optimization momentum, which is more than just smoothing the group mean. We will clarify this interpretation.
>
> ## 6. Replay Design and Entropy Weighting Trade-offs (W4)
> The replay is intentionally **minimal** to limit bias and preserve diversity. We store at most one "best" trajectory and use it **only when all $G$ current samples fail**. In this failure-only scenario, there is no successful diversity to lose. Without replay, we would update from failures only or skip the prompt. With replay, the batch statistics and stylistic diversity are still primarily driven by the $G-1$ freshly sampled, on-policy trajectories. Appendix C.2 bounds the bias introduced by the mixed gradient.
>
> Finally, the ablation shows the gain of each component independently. The entropy weighting component, though contributing a smaller gain than EMA, still offers a **positive improvement** compared to GRPO.
>
>
> We believe these clarifications and new experiments strengthen the paper and make the contributions of MAPO clearer.

---

### Official Review · Reviewer_pewJ · 2025-10-31

**Soundness:** 2
**Presentation:** 3
**Contribution:** 2
**Rating:** 4
**Confidence:** 4

**Summary:**

In this paper, the authors propose a novel critic-free policy-gradient-based reinforcement learning algorithm called Momentum-Aware Policy Optimization (MAPO). The proposed method comprises three core components: (1) a momentum-based approach for advantage estimation, (2) a confidence-based prioritized replay mechanism for historical trajectory reuse, and (3) an advantage reweighting scheme that emphasizes tokens with increasing entropy.

The proposed method demonstrates several improvements over baseline approaches such as GRPO and DAPO. The ablation studies are well-executed, and the theoretical analysis of each component is well-structured.

However, from my perspective, the motivation underlying each component lacks sufficient conviction. Additionally, the experimental evaluation is limited to a single mathematics-related dataset, leaving the validity and scalability of the proposed method yet to be fully established.

**Strengths:**

1. Despite the complexity of the notation, the presentation remains clear and accessible. Each component is introduced in a well-organized manner, avoiding potential confusion.

2. The ablation studies are comprehensive. The experiments convincingly demonstrate that each proposed component contributes to enhancing the baseline algorithm's performance.

3. Appendix C provides theoretical analysis supporting the methods presented in the main text. The proof sketches are easy to follow and closely aligned with the proposed methodologies.

**Weaknesses:**

1. Regarding the proposed "Momentum Advantage" component, the authors argue that the motivation is to address vanishing advantages when batch rewards are homogeneous. Personally, I have serious doubts about this motivation. Compared with traditional REINFORCE, the vanishing of homogeneous rewards is precisely the key to GRPO's success [1]. Similarly, DAPO discards homogeneous batched samples, thereby improving training efficiency. From the perspective of policy gradient methods, the objective is to emphasize favorable trajectories (with positive advantages) and suppress unfavorable ones (with negative advantages). It remains unclear why samples with zero advantages should be utilized, as this appears to contradict the fundamental principle of policy gradient-based methods. Furthermore, the introduced momentum mechanism results in non-zero mean advantages within each update batch, which, in my experience, may lead to faster entropy collapse or entropy blow-up.

2. The motivation underlying the proposed "Experience Replay" component is also flawed from my perspective. It is designed to address prompts that initially yield good responses but fail to do so after one epoch of training. As the policy's capability continues to improve, how is such deterioration even possible? I suspect that the activation of this module would be extremely rare during the training process of any model. Moreover, the replayed samples are approximately one epoch old, rendering them highly off-policy. Incorporating such samples into the update batch may result in excessively large KL penalties.

3. The motivation behind "entropy-guided advantage reweighting" remains questionable. Why should tokens exhibiting "higher-than-usual uncertainty" receive greater emphasis?


[1]. Xiong et.al, A Minimalist Approach to LLM Reasoning: from Rejection Sampling to Reinforce.

**Questions:**

1. Could the authors provide a more detailed explanation of the motivation underlying each proposed component?

2. The training curves of DAPO appear unusual compared with those of GRPO in Figure 3. Is the comparison fully fair? Did the authors strictly adhere to the experimental settings reported in the original DAPO paper?

3. Could the authors provide additional experimental results on other domains, such as coding or general reasoning tasks?

---

> ### Author Response · Authors · 2025-11-23
>
> We sincerely thank the reviewer for the careful reading and the thoughtful questions. We value the opportunity to clarify the motivation behind each MAPO component and the fairness of our comparisons. Below, we respond to each point in turn.
>
> ## 1. Momentum Advantage: Motivation and Effect (Q1, W1)
> We appreciate the reviewer connecting our work to the foundational insights of GRPO and [1] (Xiong et al.). We agree that discarding homogeneous failures is beneficial, as forcing updates on pure noise is detrimental. However, our motivation addresses a complementary issue specific to RLVR: the loss of signal from homogeneous successes. When a policy consistently solves a previously difficult prompt (success prob $b^{(k)} \ll 1$), standard GRPO's within-batch normalization drives the advantage $A_i^{(k)}$ to zero because rewards are uniform ($r_i \approx \mu$). This effectively stops learning despite significant temporal improvement.
>
> Recent literature [2, 3] emphasizes that negative reinforcement is crucial for maintaining exploration. MAPO’s Momentum Baseline rescues this signal by maintaining a per-prompt EMA baseline $b_i^{(k)}$ and its drift $\Delta b_i^{(k)}$, capturing the cross-iteration trend. When within-batch rewards are homogeneous, $A_i^{(k)}$ is small. However, if the policy has improved ($\Delta b_i^{(k)} > 0$), MAPO’s signal $A_i^{(k)} + \eta\ \Delta b_i^{(k)}$ remains positive, rewarding the model for outperforming its historical expectation. Thus, we are not using "zero-advantage samples" but injecting controlled temporal trend information where batch-normalization degenerates. This breaks the per-batch zero-mean property but aligns with Policy Gradient principles using a historical baseline. Strict KL regularization ensures stability, and our experiments (Fig. 3, 4b) show smooth convergence.
>
> [1] Xiong W, Yao J, Xu Y, et al. A minimalist approach to llm reasoning: from rejection sampling to reinforce[J]. arXiv preprint arXiv:2504.11343, 2025.
>
> [2] Zhu X, Xia M, Wei Z, et al. The surprising effectiveness of negative reinforcement in llm reasoning, 2025[J]. URL https://arxiv. org/abs/2506.01347.
>
> [3] Nan G, Chen S, Huang J, et al. Ngrpo: Negative-enhanced group relative policy optimization[J]. arXiv preprint arXiv:2509.18851, 2025.
> ## 2. Experience Replay: Motivation and Frequency (Q1, W2)
> We thank the reviewer for the detailed critique of the replay component. The concern “if the policy improves, how can it fail on solved prompts?” assumes monotonic improvement. In RLVR, the objective couples many prompts, and the process is highly non-stationary. Gradient updates from other prompts and the KL constraint can temporarily reduce success margins, leading to occasional "all-fail" groups on hard problems.
>
> Our replay mechanism is conservative and targeted: we store at most one verifier-approved success per prompt and consult the buffer only when all $G$ on-policy samples fail. In this case, using only failures discards proof that the policy can solve the problem. Mixing in a single stored success keeps $G-1$ trajectories freshly sampled, so the batch remains predominantly on-policy. Empirically, replay reduces regressions and smooths learning curves. Appendix C.2 bounds the off-policy bias by a constant times the trigger probability $\lambda_k$, which decays over training, ensuring a controlled, transient bias.
>
> ## 3. Entropy-Guided Advantage Reweighting (Q1, W3)
> We thank the reviewer for questioning this motivation.
> The key design is using **entropy deviation**, not raw entropy. We define
> $$\Delta H_{i,t}^{(k)} = H_{i,t}^{(k)} - \bar H_{i,t}^{(k)}$$ and normalize it to obtain weights $w_{i,t}^{(k)}$.
> - Tokens that are always uncertain for both policies (e.g., formatting) have small $\Delta H_{i,t}^{(k)}$ and receive near-average weights, so they are not amplified.
> -  Tokens whose uncertainty is unusually high **relative to the reference**—typically critical forks or key algebraic steps—receive larger weights, while the total sequence-level advantage is preserved.
>
> We are therefore not claiming that high entropy alone identifies important reasoning tokens; rather, we use relative entropy on successful trajectories to locate positions where the current policy remains more uncertain than a strong baseline and concentrate the gradient there without increasing its overall magnitude. As the work by Wang et al. [1] found, only a minority of high-entropy tokens are critical to the reasoning process. These tokens act as **critical forks** that dictate the model's capacity to explore various reasoning paths. Our qualitative case studies indicate that these high-weight tokens indeed align with critical decision points, and we will emphasize this “relative entropy” interpretation more clearly in the revised text.
>
> [1] Wang S, Yu L, Gao C, et al. Beyond the 80/20 rule: High-entropy minority tokens drive effective reinforcement learning for llm reasoning[J]. arXiv preprint arXiv:2506.01939, 2025.

---

> > ### Author Response · Authors · 2025-11-23
> >
> > ## 4. Strict Adherence to Experimental Settings (Q2)
> > We thank the reviewer for this important question. To ensure a strictly fair comparison, we enforced a unified experimental setting for all methods. We used identical model architectures, datasets, group sizes, rollout temperatures, and learning rate schedules across GRPO, DAPO, and MAPO. Additionally, the DAPO implementation strictly follows the hyperparameters reported in the original paper (including batch size and update steps) without introducing any MAPO-specific modifications.
> >
> > The apparent difference in training curves (Figure 3) arises mainly because we plot all methods under the **same** decoding and evaluation protocol (Pass@1 / Pass@N over the same benchmarks), whereas the original DAPO paper includes some settings tailored to specific datasets. Under this rigorous unified protocol, DAPO still consistently outperforms the GRPO baseline, confirming the correctness of our implementation. We will add these implementation details to the revision for clarity.
> >
> > ## 5. Other Domains and Generality (Q3)
> >
> > We thank the reviewer for suggesting the extension to other domains. We agree that evaluating generality is important. However, we prioritized mathematical reasoning in this work for two primary reasons:
> > 1. Benchmarking Standards \& Fair Comparison: Mathematical reasoning has become the de facto standard testbed for recent Policy Optimization (PO) research (e.g., GRPO, DAPO). By strictly adhering to these established benchmarks, we ensure that our comparisons are rigorous and that gains are due to the algorithmic improvements of MAPO rather than domain-specific discrepancies or unaligned baselines.
> > 2. Objective Verifiability: Our focus is on RL with Verifiable Rewards (RLVR). Math tasks provide objective, binary (0/1) ground-truth signals, which are essential for cleanly validating the effectiveness of the verifier mechanism without the noise introduced by heuristic reward models used in general reasoning.
> >
> > We are actually actively extending MAPO to tasks from other domains. We consider it a priority for our immediate next steps.

---

### Official Review · Reviewer_YkhJ · 2025-10-31

**Soundness:** 3
**Presentation:** 3
**Contribution:** 2
**Rating:** 6
**Confidence:** 3

**Summary:**

This paper proposes Momentum-Aware Policy Optimization (MAPO), an extension to GRPO. The method combines a momentum group baseline to prevent gradient collapse under low reward variance, a confidence-based prioritized replay mechanism to reuse verified successes, and an entropy-weighted token update to focus learning on uncertain reasoning steps. Experiments on several mathematical reasoning benchmarks show improvements over GRPO and DAPO in Pass@1 and Pass@N, with ablation studies indicating complementary effects among the three components.

**Strengths:**

1. The paper targets an important weakness of GRPO in realistic low-variance reward settings and provides an elegant baseline mechanism with quantifiable variance reduction.
2. The theoretical analysis is detailed, including finite group and variance decomposition results.
3. Experimental results are clearly demonstrate sample efficiency and learning stability gains.

**Weaknesses:**

1. The integration between the three proposed components is not fully clear. The paper could discuss a unifying motivation or design principle, since the “momentum‑aware” name primarily describes only the first component.
2. Some evaluation benchmarks, particularly AIME24/25, are small in size, which limits the statistical reliability of reported improvements. Confidence intervals or significance tests would make the results more convincing.
3. To better position MAPO among recent methods, the related work discussion should be expanded to include both replay-based optimization (e.g., RePO, RLEP) and recent work on token reweighting (e.g., 'Beyond the 80/20 Rule...'; 'Do Not Let Low-Probability Tokens...').
4. The off‑policy nature of replayed samples is a potential concern. Appendix C.2 provides a theoretical bound, but the main text should summarize this clearly to justify the design.
5. The hyperparameters α and η are fixed throughout. A short sensitivity analysis would help confirm that the method is not overly dependent on these values.
6. In Table 1, MAPO’s improvements over baselines are sometimes small or inconsistent across datasets and metrics, and clarification on these fluctuations would help interpret the results.

**Questions:**

1. How sensitive is MAPO to EMA decay α and momentum weight η? Could a high α lead to stale baselines?
2. Is there a deeper conceptual connection among the three mechanisms beyond their empirical complementarity?
Strengths

---

> ### Author Response · Authors · 2025-11-23
>
> We sincerely thank the reviewer for the careful reading, and the very helpful questions and suggestions.
>
> ## 1. Sensitivity of $\alpha$ and $\eta$, and stale baselines  (Q1 \& W5).
> We thank the reviewer for highlighting this issue.
> The momentum group baseline and advantage are defined in Section 3.1. For any $\alpha\in(0,1)$, the EMA baseline converges geometrically to the per-prompt success probability while preserving the $O(G^{-1})$ finite-group variance scaling of GRPO; the $\eta$ term acts as a small control variate, with bias linear in $\eta$ when $\eta$ is small.
>
> To quantify sensitivity, we conducted an ablation with the results given below.
>
> | Setting             | AIME24 Pass@1 | AIME24 Pass@32 | AMC23 Pass@1 | AMC23 Pass@32 |
> |---------------------|---------------|----------------|--------------|---------------|
> | GRPO (1.0, 0.0)     | 20.0          | 53.3           | 62.5         | 87.5          |
> | MAPO (0.5, 0.1)     | 23.3          | 56.7           | 60.0         | 85.0          |
> | MAPO (0.7, 0.1)     | 26.7          | 56.7           | 67.5         | 87.5          |
> | **MAPO (0.9, 0.1)** | **30.0**      | **66.7**       | **67.5**     | **95.0**      |
> | MAPO (0.9, 0.2)     | 23.3          | 63.3           | 67.5         | 92.5          |
> | MAPO (0.9, 0.3)     | 23.3          | 60.0           | 67.5         | 87.5          |
>
> We observe that
> 1. smaller $\alpha$ (0.5, 0.7) gives stronger dependence on past baselines and slower adaptation, degrading performance;
> 2. larger $\eta$ (0.2, 0.3) makes the drift term dominate when $\Delta b_i^{(k)}$ fluctuates, increasing bias and hurting AIME24;
> 3. $(\alpha,\eta)=(0.9,0.1)$ provides the best trade-off and consistently improves over GRPO.
>
> Thus, in our setting, “stale baselines” arise for too small $\alpha$, not for large $\alpha$. A very large $\alpha$ (e.g., $\alpha=1$) instead corresponds to a purely batch-based baseline (GRPO), which is highly reactive but does not benefit from temporal smoothing. The ablation indicates that MAPO’s gains are robust within a reasonable range of $(\alpha,\eta)$ and that the default choice $(0.9,0.1)$ avoids both extremes.
>
> ## 2. Conceptual integration and “momentum-aware” design (Q2, W1).
>
> We thank the reviewer for asking for a unified perspective.  MAPO is not three independent tricks, but a single modified GRPO gradient estimator where all components jointly shape the advantage signal
>
> $\hat{\nabla } L_{ MAPO} ( {\theta }_k)   ∝   [( 1-\lambda _k ) $
>
> $E_{\tau \sim \pi_{\theta_k}} ( \sum_t w_{i,t}^{(k)} (A_i^{(k)} + \eta \Delta b_i^{(k)}) \nabla_\theta log \pi_{\theta_k}(y_{i,t} | y_{i,<t}, x_i) ) ] +\lambda_k  E_{\tau \sim {B_k}}  ( \sum_t w_{i,t}^{(k)} (A_i^{(k)} + \eta \Delta b_i^{(k)}) \nabla_\theta log \pi_{\theta_k}(y_{i,t} | y_{i,<t}, x_i)  )$
>
> - The momentum-aware per-prompt baseline ($\Delta b_i^{(k)}$) stabilizes advantages over time;
> - Replay (with $\lambda_k$) injects verified successes only on failure, preserving reward integrity while reducing variance;
> - Entropy-based token weights ($w_{i,t}^{(k)}$) redistribute the scalar advantage across positions without altering its total.
>
> ## 3. Small benchmarks and fluctuations in Table 1 (W2 \& 6).
>
> We thank the reviewer for emphasizing this caveat.AIME24/25 and AMC23 are 30–40 problems, so ±1–2 solved items cause noticeable Pass@1 fluctuations.   To mitigate this, we report average performance across four benchmarks (including larger MATH500). MAPO consistently outperforms GRPO/DAPO—especially Qwen2.5-Math-7B—under identical single-run protocols (model, decoding, budget), and we will clarify this and the evaluation protocol in the revised version.
>
> ## 4. Relation to replay-based and token-reweighting methods (W3)
> We thank the reviewer for these helpful pointers. The current draft cites RLEP and “Beyond the 80/20 Rule…”; we’ll add RePO and “Do Not Let Low-Probability Tokens…” and sharpen comparisons. Conceptually, MAPO differs in two aspects:
> 1. Its replay buffer keeps only the most recent successful sample per prompt (overwriting any previous one), and replays it only when the current rollout group yields no success;
> 2. Its entropy weighting uses normalized entropy deviations while preserving the sequence-level advantage.
> We view MAPO as complementary and will clarify this in the revised related work.
>
> ## 5. Off-policy replay and  clarification (W4)
> We thank the reviewer. Appendix C.2 proves that the gradient bias is bounded by $\lambda_k B$($\bigl\|\mathbb{E}[\hat\nabla L_{\mathrm{mix}}(\theta)] - \mathbb{E}[\hat\nabla L_{\mathrm{on}}(\theta)]\bigr\|
> \le \lambda_k B$), where $\lambda_k$ is the probability of triggering replay. Empirically, $\lambda_k$ is significant only in early training and decays rapidly as on-policy success rates improve. Thus, replay acts as a temporary safety net with diminishing bias, rather than introducing persistent off-policy drift. We will incorporate a summary of this bound into the main text.

---

### Official Review · Reviewer_LU4f · 2025-11-02

**Soundness:** 3
**Presentation:** 3
**Contribution:** 3
**Rating:** 6
**Confidence:** 4

**Summary:**

The paper introduces a novel framework that enhances Large Language Model reasoning through Reinforcement Learning with Verifiable Rewards (RLVR). It tackles two key limitations of group-based methods like GRPO: vanishing learning signals due to low reward variance and suboptimal token-level credit assignment. MAPO's solution integrates three core components: a Momentum Group Baseline for stable learning signals, a Confidence-Based Prioritized Replay for sample efficiency, and Entropy-Weighted Token Updates for precise gradient focus. Extensive evaluations on mathematical reasoning benchmarks demonstrate MAPO's superiority over GRPO and DAPO in both Pass@1 and Best-of-N accuracy, with faster convergence and improved training stability.

**Strengths:**

1. The Momentum Group Baseline stands out as a significant contribution. It elegantly resolves a fundamental issue in group-based policy optimization by employing an Exponential Moving Average (EMA) of historical rewards. This ensures a persistent learning signal, directly countering a primary cause of training stagnation and fostering more robust and continuous improvement.

2. The framework is highly practical, functioning as a "drop-in" enhancement to existing methods like GRPO. Its three components are complementary yet conceptually independent, facilitating straightforward implementation and enabling clear ablation studies to understand the contribution of each part.

**Weaknesses:**

1. The design of the prioritized replay buffer appears somewhat conservative, as it stores only the single highest-confidence successful trajectory per prompt. This strategy may inadvertently limit the diversity of positive examples during training, potentially constraining the explored solution space. There is a risk of the model overfitting to a specific style of successful trajectory, particularly for problems that admit multiple valid solution paths.

2. While the results on mathematical reasoning benchmarks are compelling, the empirical evaluation is confined to this single domain. The paper does not demonstrate the framework's efficacy on other prominent RLVR tasks, such as code generation or symbolic reasoning. This leaves its generalizability and performance across a broader range of domains as an open question.

**Questions:**

The hyperparameters for the momentum mechanism (EMA decay α and momentum coefficient η) are crucial to the method's performance. How sensitive are the results to the specific values chosen (0.9 and 0.1, respectively), and is there a rationale or empirical evidence for these choices? Could an adaptive scheduling mechanism for these parameters further improve performance across different training phases or tasks?

The entropy-weighting mechanism aims to focus learning on "critical decision points," which is assumed to correlate with high token-level entropy. However, could this approach inadvertently amplify the gradient for tokens that are highly uncertain but semantically unimportant (e.g., transitional phrases or stylistic variations)? How does the method ensure that high entropy reliably indicates a critically reasoning-relevant token?

---

> ### Author Response · Authors · 2025-11-23
>
> We sincerely thank the reviewer for the careful review, the questions, and the suggestions. Below, we address each of the raised points individually.
>
> ## 1 Sensitivity and rationale for the momentum hyperparameters $\alpha$ and $\eta$ (Q1)
>
> We thank the insightful question.
> The momentum group baseline and advantage are defined in Section 3.1. For any $\alpha\in(0,1)$, the EMA baseline converges geometrically to the per-prompt success probability while preserving the $O(G^{-1})$ finite-group variance scaling of GRPO; the $\eta$ term acts as a small control variate, with bias linear in $\eta$ when $\eta$ is small.
>
> To quantify sensitivity, we conducted an ablation with the results given below.
> | Setting             | AIME24 Pass@1 | AIME24 Pass@32 | AMC23 Pass@1 | AMC23 Pass@32 |
> |---------------------|---------------|----------------|--------------|---------------|
> | GRPO (1.0, 0.0)     | 20.0          | 53.3           | 62.5         | 87.5          |
> | MAPO (0.5, 0.1)     | 23.3          | 56.7           | 60.0         | 85.0          |
> | MAPO (0.7, 0.1)     | 26.7          | 56.7           | 67.5         | 87.5          |
> | **MAPO (0.9, 0.1)** | **30.0**      | **66.7**       | **67.5**     | **95.0**      |
> | MAPO (0.9, 0.2)     | 23.3          | 63.3           | 67.5         | 92.5          |
> | MAPO (0.9, 0.3)     | 23.3          | 60.0           | 67.5         | 87.5          |
>
> We observe that
> 1. smaller $\alpha$ (0.5, 0.7) gives stronger dependence on past baselines and slower adaptation, degrading performance;
> 2. larger $\eta$ (0.2, 0.3) makes the drift term dominate, and when $\Delta b_i^{(k)}$ fluctuates, it may increase bias;
> 3. $(\alpha,\eta)$=(0.9,0.1) provides the best trade-off and consistently improves over GRPO.
>
> We agree that adaptive scheduling is a promising direction. One can view $1/(1-\alpha)$ as an effective memory length and $\eta$ as the drift strength: larger $\eta$ and shorter memory early on, followed by increasing $\alpha$ / decaying $\eta$ as baselines stabilize, is a natural strategy. To keep MAPO simple and drop-in compatible, we fix $(\alpha,\eta)$; exploring adaptive schedules is left for future work.
>
> ## 2. Entropy-weighted token updates and semantically unimportant tokens (Q2)
> We appreciate this thoughtful concern.
> The entropy weighting in Section 3.3 (Eq.(6)) is based on **entropy deviation** $\Delta H_{i,t}^{(k)} = H_{i,t}^{(k)} - \bar H_{i,t}^{(k)}$, not raw entropy. $\Delta H_{i,t}^{(k)}$ is then normalized along the sequence into weights $w_{i,t}^{(k)}$.
> This means
> - Tokens that are high-entropy for both policies (e.g., format variations) have small $\Delta H_{i, t}^{(k)}$ and receive near-average weights.
> - Tokens whose uncertainty is unusually high relative to the reference (i.e., where uncertainty grew the most) receive larger weights, without changing the total sequence-level signal $\left(\sum_t \tilde{A}_{i, t}^{(k)}=A_i^{(k)}\right)$.
>
> This aligns with observations that only a minority of high-entropy tokens act as "critical forks" for reasoning [1]. Our case studies (Section 4.5) show that high-weight tokens concentrate at decision points rather than purely decorative words. We will emphasize this relative entropy viewpoint more clearly in the main text.
>
> [1] Wang S, Yu L, Gao C, et al. Beyond the 80/20 rule: High-entropy minority tokens drive effective reinforcement learning for llm reasoning[J]. arXiv preprint arXiv:2506.01939, 2025.
> ## 3. Prioritized replay: single best trajectory and diversity (W1)
> We thank the reviewer for the insightful remark. Our replay buffer is intentionally **minimal and tailored to RLVR**: we store only **one** verifier-approved, high-confidence success per prompt, and consult it only when all $G$ on-policy samples fail.
> - The replay provides the necessary successful signal when the current batch consists only of failures.
> - Exploration is largely maintained since $G-1$ trajectories remain freshly sampled.
> - In ablations, this mechanism consistently improves stability and sample efficiency without style collapse, offering a good simplicity–performance trade-off.
>
> ## 4. Domain coverage beyond mathematical reasoning (W2)
> We thank the reviewer for raising the generality question. We focus on mathematical reasoning because it is a central RLVR application with strong verifiers and widely used baselines, enabling a stringent and feasible comparison.
>
> Algorithmically, MAPO only assumes group-based sampling with verifiable rewards.
> - The momentum baseline uses per-prompt Bernoulli rewards.
> - Replay stores verifier-approved successes.
> - Entropy weighting operates on token probabilities.
>
> These assumptions hold equally for tasks like code execution and symbolic reasoning. The consistent improvements over strong baselines across four math benchmarks and three model sizes suggest MAPO is useful in a demanding setting, and extending it to other RLVR domains is a natural next step rather than a limitation of the approach.

---

### Note · Authors · 2026-01-26

I have read and agree with the venue's withdrawal policy on behalf of myself and my co-authors.

---

### Meta-Review · Area_Chair_84jb · 2026-01-06

**Summary:**

The scores given by the reviewers for this paper is at the borderline, with two positives 6,6 and two negatives 4,2.
I think the proposed method has a good motivation, and the designed group EMA is a reasonable estimate of the baseline. However, the concerns brought up by the reviewers are valid and not fully addressed. Therefore, I recommend rejection.

**Reviewer Concerns:**

Outstanding concerns:

1. methodology itself is incremental
2. evaluation is on limited dataset and insufficient (not addressed in rebuttal)
3. improvements are inconsistent (verifiied in the result table of the paper)
4. the added table shows that result varies largely with different hyperparameters

**Reviewer Scores:**

The scores are 6642, at the borderline, for the concerns listed above, the rebuttal doesn't address enough

---

### Decision · Program_Chairs · 2026-01-26

Reject